

# Synchronization frequency analysis and stochastic simulation of multisite flood flows based on the complicated vine-copula structure

Xinting Yu, Yuxue Guo, Siwei Chen, Haiting Gu, Yue-Ping Xu[*]

Institute of Water Science and Engineering, Civil Engineering and Architecture, Zhejiang University, Hangzhou, 310058, China

*Correspondence to*: Yue-Ping Xu (yuepingxu@zju.edu.cn)

**Abstract:** Accurately modeling and predicting flood flows across multiple sites within a watershed presents significant challenges due to potential issues of insufficient accuracy and excessive computational demands in existing methodologies. In response to these challenges, this study introduces a novel approach centered around the use of vine copula models, termed RDV-Copula (Reduced-dimension vine copula construction approach). The core of this methodology lies in its ability to integrate and extract complex data information before constructing the copula function, thus preserving the intricate spatial-temporal connections among multiple sites while substantially reducing the vine copula's complexity. This study performs a synchronization frequency analysis using the devised copula models, offering valuable insights into flood encounter probabilities. Additionally, the innovative approach undergoes validation by comparison with three benchmark models, which vary in dimensions and nature of variable interactions. Furthermore, the study conducts stochastic simulations, exploring both unconditional and conditional scenarios across different vine copula models. Applied in the Shifeng Creek watershed, China, the findings reveal that vine copula models are superior in capturing complex variable relationships, demonstrating significant spatial interconnectivity crucial for flood risk prediction in heavy rainfall events. Interestingly, the study observes that expanding the model's dimensions does not inherently enhance simulation precision. The RDV-Copula method not only captures comprehensive information effectively but also simplifies the vine copula model by reducing its dimensionality and complexity. This study contributes to the field of hydrology by offering a refined method for analyzing and simulating multisite flood flows.



## 1 Introduction

Floods are the most frequent natural disaster, inflicting substantial economic losses, environmental degradation and human casualties (Teng et al., 2017). As is reported by Centre for Research on the Epidemiology of Disasters (CRED), floods represented 45.6% of worldwide natural disasters in 2022, affecting an average of 57.1 million people annually (CRED,2023). The data also indicated a 4.76% increase in flood occurrences in 2022 compared to the annual average from 2002 to 2021(CRED,2023). Therefore, it is very meaningful and essential to analyze flooding and achieve flood risk control. At the watershed scale, flood risk is primarily influenced by rainfall patterns and interconnections among sub-watersheds. Large floods often result from the amalgamation of floods from multiple sub-watersheds (Prohaska and Ilic, 2010). Concurrent flood events cause runoff from various sources to merge, forming large floods that pose threats to downstream regions. As a result, analyzing the runoff at various sites not only provides a better understanding of the flood characteristics within the watershed, but also contributes to the development of flood control programs to avoid flood risks.

There are currently many techniques for analyzing hydrological variables. Common univariate methods include statistical analyses such as frequency analysis (Stedinger et al., 1993), extreme value theory (Coles, 2001), and time series analysis methods like the Autoregressive Integrated Moving Average (ARIMA) model (Box et al., 2013). However, univariate analyses often fall short in accurately estimating the risks associated with extreme events due to their inability to account for the interdependence of variables (Khan et al., 2023). This oversight can lead to significant underestimation or overestimation of risks, particularly given the inherent relationships among variables within a catchment. To address the complexity of these relationships across multiple variables, researchers have turned to multivariate analysis techniques. Methods such as Autoregressive (AR) models are utilized for analyzing temporal correlations (Box et al., 2013), while spatial relationships can be examined using techniques like geostatistical methods (Isaaks and Srivastava, 1989), spatial regression models (Bekker and Wansbeek, 2001), Copula functions (Sklar, 1959) and Bayesian hierarchical models (Gelman et al., 2013). However, these methods have their limitations. AR models, while effective for temporal analysis, do not account for spatial dependencies. Geostatistical methods and spatial regression models focus primarily on spatial relationships but may struggle with temporal dynamics. Bayesian hierarchical models can handle complex dependencies but often involve high computational demands and require



substantial prior information. In contrast, copula functions offer substantial advantages when dealing
with multivariate spatial-temporal relationships. They provide a flexible framework for modeling
dependencies between variables without assuming a specific marginal distribution, allowing for a more
accurate representation of complex interdependencies. Later adopted in hydrology by De Michele and
Salvadori (2003), copula functions link multidimensional probability distribution functions to their one-
dimensional margins, preserving both the dependence structure and the distinct distribution
characteristics of random variables (Tosunoglu et al., 2020). Copula function is widely applied in
hydrological fields, including the joint frequency analysis (Liu et al., 2018; Zhang et al., 2021), water
resources management (Gao et al., 2018; Nazeri Tahroudi et al., 2022), wetness-dryness encountering
(Wang et al., 2022; Zhang et al., 2023), flood risk assessment (Li et al., 2022; Tosunoglu et al., 2020;
Zhong et al., 2021) , water quality analysis (Yu et al., 2020; Yu and Zhang, 2021),precipitation model
(Gao et al., 2020; Nazeri Tahroudi et al., 2023; Tahroudi et al., 2022) and so on.

Despite the broad application of conventional copula functions to create joint distributions for

multiple variables, their capacity to accurately represent high-dimensional realities is constrained. This
limitation arises from their reliance on a single parameter to describe correlations and a simplistic
approach to model the dependence structure between variables (Aas et al., 2009; Daneshkhah et al., 2016).
To overcome these limitations, Bedford and Cooke (2002) proposed a reliable way called Vine Copula
to construct complex multivariate models with high dependency. Vine copula construction relies
exclusively on the principle of breaking down the complete multivariate density into a series or simple,
foundational components through conditional independence or pair-copula constructs. There are two
main types of vine structures: C-Vine and D-Vine (Brechmann and Schepsmeier, 2013). The former
presents star-shaped configurations, while the latter displays path-like structures, providing enhanced
flexibility in constructing the joint distribution of multiple variables by enabling the use of different types
of bivariate copulas for each pair, thus accommodating a diverse range of dependency structures (Aas et
al., 2009; Çekin et al., 2020).

Vine copulas are increasingly applied in hydrological studies to model complex relationships among

multiple variables. For instance, Ahn (2021) developed a D-vine copula-based model to estimate flows
in catchments with limited or partial gauging, focusing on the temporal relationship of runoff at a specific
site. This model employed a six-dimensional copula structure centered around annual runoff, using



conditional simulation to compensate for missing data. Wang et al. (2022) explored the joint distribution
of multi-inflows to assess wetness-dryness conditions, highlighting spatial interconnections across three
water systems but ignoring the temporal influences within each system on the overall assessment. Unlike
the above studies, Xu et al. (2022) developed a stepwise and dynamic C-vine copula-based conditional
model (SDCVC) to incorporate the non-stationarity into a monthly streamflow prediction. This model
synthesizes the temporal and spatial relationships at multiple sites, developing a four-dimensional C-vine
copula for dual-site monthly streamflow forecasts. The term "four dimensions" relates to the categories
of variables involved, such as rainfall, downstream station streamflow, among others. Integrating
temporal and spatial relationships in copula construction allows for a more comprehensive data inclusion,
facilitating enhanced modeling of complex inter-variable relationships. However, challenges arise as the
number of sites or the analysis period extends, leading to increased complexity and dimensionality of the
copula function. This complexity can complicate the copula's structure determination, inflate
computational demands during parameter fitting, and potentially diminish the accuracy of stochastic
simulations. To bridge this gap, this study aims to propose a new approach to achieve dimensionality
reduction while ensuring the complete access of spatial-temporal relationships for multiple sites. The
primary focus is to filter effective information to fully incorporate runoff data from each site and mitigate
the complexity of the vine copula function, thereby preventing poor model fitting due to increased
computational effort.
Moreover, understanding the spatial and temporal relationships of runoff across multiple sites within
a catchment is essential for effective flood control and water resources management. Synchronization
probability analysis and stochastic simulation of streamflow sequences play a pivotal role in these
processes (Chen et al., 2015). The terminology used to describe the encounter situations of wetness and
dryness varies; an asynchronous event refers to a scenario where such encounters do not occur
simultaneously, whereas both wetness-wetness and dryness-dryness encounters are considered
synchronous events. These encounters exist not only in diversion projects and multi-source water supply
systems, but also in main streams and tributaries at a watershed scale. They offer invaluable insights into
the spatial and temporal distribution of water resources, aiding in the preparation for anticipated future
events (Szilagyi et al., 2006). Copula-based simulation was first discussed in the study of Bedford and
Cooke (2001;2002). Subsequently, as more studies have been conducted, copula-based modeling and



simulation models for hydrological variables have demonstrated high performance (Gao et al., 2021;
Huang et al., 2018; Tahroudi et al., 2022). Utilizing stochastic simulation to generate sets of runoff
sequences from multiple sites not only allows for a more progressive test of the effectiveness of the vine
copula function in fitting the relationship, but also provides a data base for flood control scheduling in
making decisions.

The basic task of this study is to construct the relationship functions of runoff across multiple sites

within a catchment using the vine copula. By leveraging the copula model, the frequency of flood
encounters for multiple runoffs is calculated to further analyze the intrinsic spatial and temporal
relationship characteristics. Addressing the challenge of dimensionality disaster caused by excessive
variables, this study proposes a novel approach to reduce the dimensionality by filtering the effective
information under the premise of fully incorporating the runoff information from each site. This approach
makes it possible to access the spatial and temporal relationships of runoff from multiple sites in the
catchment more accurately and efficiently. In addition, more reality-oriented simulation results can be
obtained, which provide statistical support for flood control and scheduling decision-making.

This paper is structured as follows: Section 2 outlines the proposed methodology's framework.

Section 3 presents the application of this methodology through a case study. The results are detailed in
Section 4, while Section 5 provides a thorough analysis and discussion of the results. Finally, Section 6
concludes the paper by summarizing the principal conclusions.
**2    Methodology**
The framework of this study is shown in Figure 1. This Section focuses on constructing and applying
multivariate joint distribution functions based on the vine copula function. It is divided into two cases:
one considering only spatial relations and the other combining spatial and temporal relations. Utilizing
the data characteristics, it describes how to build a vine copula function based on multiple variables and
details the processes of synchronization frequency analysis and stochastic simulation with the
constructed vine copula function. Additionally, it presents a new approach called the reduced-dimension
vine copula (RDV-Copula).



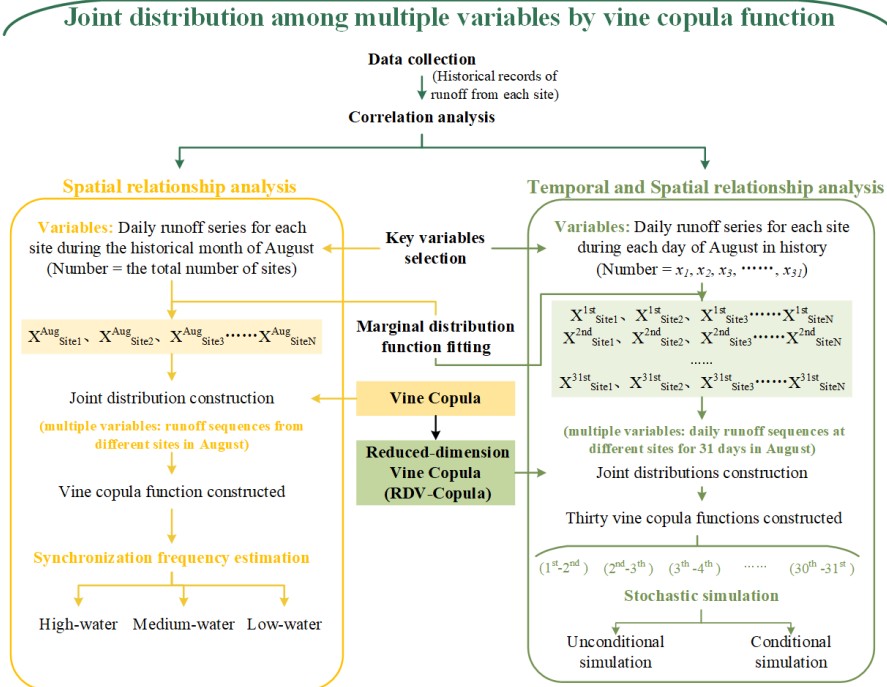

**Figure 1. Framework of proposed methodology**

## 2.1 Joint distribution of multiple variables

Before identifying the dependence relationships among multi-variables, their correlations need to be analyzed and judged. Kendall's correlation coefficient, a nonparametric statistic, serves to measure the correlation degree between two variables, making it suitable for nonlinear relationships and categorical variables. In this study, vine copula functions are constructed to achieve synchronization frequency and stochastic simulation of multiple streamflow sequences. To more accurately simulate the temporal and spatial relationships, the correlations among multi-site streamflow series are determined by calculating the Kendall correlation coefficients.

### 2.1.1 Marginal distribution function

To build the dependence structure of hydrological variables using copulas, it is essential to determine the marginal distribution of each variable first. Given that the marginal distribution function for each characteristic variable is not predetermined and the skewness of their probability distributions varies (Zhong et al., 2021), it becomes crucial to consider multiple marginal distribution functions as candidates.



In this study, a comprehensive comparison is conducted among 12 commonly utilized distributions
(Tosunoğlu, 2018), including Gamma distribution (gamma), Exponential distribution (exp), Pearson-III
distribution (p3), Generalized extreme value distribution (gev), Inverse gaussian distribution (invgauss),
Normal distribution (norm), Logistic distribution (logis), Log-normal distribution (lnorm), Log-logistic
distribution (llogis), Generalized pareto distribution (gpd), Weibull distribution (weibull) and Gumbel
distribution (gumbel). According to the goodness-of-fit test and AIC minimum criterion, the optimal
distribution functions are selected as the marginal functions of the characteristic variables. The specific
details of different distributions, such as the probability distribution function and the respective
parameters, are displayed in Appendix A.
**2.1.2 Vine copula function theory**
Copula functions, first introduced in 1959, represent a multivariate joint probability distribution function
within the unit square [0, 1], featuring uniform marginal distributions. According to Sklar's theorem
(Sklar, 1959), for a multivariate random variable $x_1, x_2, x_3, \cdots, x_d$, there exist marginal distributions
$u_1 = f_1(x_1),\ u_2 = f_2(x_2),\ u_3 = f_3(x_3), \cdots, u_d = f_d(x_d)$ and joint distribution $f(x_1, x_2, x_3, \cdots, x_d)$,
then there exists a copula function $C_\theta$ such that
$$f(x_1, x_2, x_3, \cdots, x_d) = C_\theta[f_1(x_1), f_2(x_2), \cdots, f_d(x_d)] = C_\theta(u_1, u_2, \cdots, u_d) \qquad (1)$$
If $f_1(x_1),\ f_2(x_2), \cdots,\ f_d(x_d)$ are continuous functions, then $C$ is unique. $\theta$ represents an
explicit parameter to the function.
The multivariate conditional density function can be represented as:
$$f(x|v) = C_{xv_j|v_{-j}}\left(F(x|v_{-j}), F(v_j|v_{-j})\right) f(x|v_{-j}) \qquad (2)$$
where $v_j$ denotes a component of the n-dimensional vector $v$, while $v_{-j}$ denotes the (n-1)-dimensional
vector with $v_j$ removed.
The term $f(x|v)$ in each conditional density function can be denoted as:
$$F(x|v) = \frac{\partial C_{xv_j|v_{-j}}\left(F(x|v_{-j}), F(v_j|v_{-j})\right)}{F(v_j|v_{-j})} \qquad (3)$$
The copula function, essentially, acts as a transformation function that connects the joint distribution
of multiple variables to the marginal distributions. There are a number of alternative copula families that
can be selected for the construction of modeling dependence, such as Gaussian copula, t-copula, Clayton
copula, Gumbel copula, Frank copula and so on. However, the construction of high-dimensional copula



functions is often constrained by parameter limitations and computationally demanding. Bedford and
Cooke (2002) introduced a more advanced and flexible alternative method of constructing the
dependence structure called Vine Copula. Also later called pair-copula construction by Aas et al. (2009),
vine copulas decompose the joint density function into a cascade of building blocks of the bivariate
copulas. Assuming that there are $d$ variables given to us, it is possible by this method to decompose the
d-dimensional joint distribution into $d(d-1)/2$ pair copulas densities. In vine copula structure, the
vine consists of a series of trees, nodes, and edges. The trees represent the layers. Each layer contains
several nodes and the connections between the nodes are called the edges. The nodes in the first tree
represent the marginal distributions of each variable. Each edge represents a pair-copula joint distribution
function of two adjacent nodes. The edges in each tree, except the last tree, are used as nodes in the next
tree. There are two subsets of regular vines in commonly use: canonical vines (C-vines) and drawable
vines (D-vines). Both types of vine-copula have their own specific way of decomposing the density
function.
C-vine is suitable for structures with a key variable that has a significant correlation with the
remaining other variables. However, in D-vine structure, each node is linked to at most two edges. The
order of dependencies between variables can be determined by one after the other. The expressions for
the $n$-dimensional joint probability density of C-vine and D-vine are shown in Equations (4) and (5).
$$f(x_1,\cdots,x_d) = \left[\prod_{j=1}^{d-1}\prod_{i=1}^{d-j} c_{j,j+1|1,\cdots,j-1}\right] \cdot \left[\prod_{k=1}^{d} f_k(x_k)\right] \text{ (C-vine)} \tag{4}$$
$$f(x_1,\cdots,x_d) = \left[\prod_{j=1}^{d-1}\prod_{i=1}^{d-j} c_{i,(i+j)|(i+1),\cdots,(i+j-1)}\right] \cdot \left[\prod_{k=1}^{d} f_k(x_k)\right] \text{ (D-vine)} \tag{5}$$
where $c(\ )$ refers to the bivariate copula with index $i$ running over the edges for each tree and index $j$
identifying the trees, $f_k(x_k)$ denotes the marginal density.
**2.2 Estimation of inflow synchronization frequency**
A distinct advantage of the copula method lies in its precision in analyzing inflow encounter probabilities
and conditional probabilities. In this study, a synchronization event is defined as the simultaneous
occurrence of inflows of similar magnitudes from multiple sites. We categorize the flow into three levels:
high, medium, and low. The frequencies associated with high-water and low-water events are set as $P_h =$
$37.5\%$ and $P_l = 62.5\%$. It is assumed that there is a generalized reservoir group scheduling system, as
shown in Figure 2. The system encompasses $N$ reservoirs and $M$ flood control cross sections.



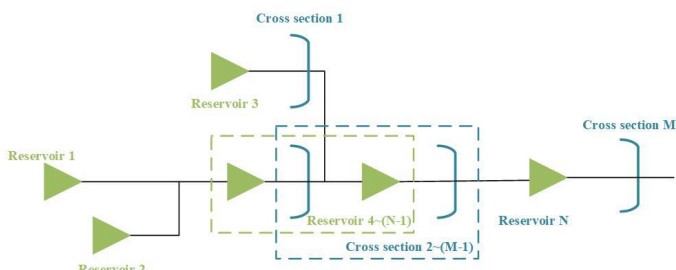


**Figure 2. Schematic diagram of the generalized system in the catchment**
We can generalize all reservoirs and cross-sections to multiple sites within the watershed system.
Each of these sites may be exposed to incoming flows when rainfall occurs. Let $X_{ph}$ and $X_{pl}$ be the
amounts of water corresponding to $P_h$ and $P_l$, respectively. $X_i > X_{ph}$ corresponds to high-water (H),
$X_i < X_{pl}$ corresponds to low-water (L), and $X_{pl} < X_i < X_{ph}$ corresponds to medium-water (M), where
$X_i$ denotes the inflow of day $i$ .
Let the inflows of the different sites be represented by $X^1, X^2, X^3, \cdots, X^{N+M}$ .
$X_{ph}^1, X_{ph}^2, X_{ph}^3, \cdots, X_{ph}^{N+M}$ represent the amounts of inflow corresponding to the high-water of these
different sites respectively. Meanwhile, $X_{pl}^1, X_{pl}^2, X_{pl}^3, \cdots, X_{pl}^{N+M}$ represent the amounts of inflow
corresponding to the low-water of these different sites respectively. The marginal distribution functions
are $u^1, u^2, u^3, \cdots, u^{N+M}$, respectively.
The number of possible inflow-state combinations increases with the number of sites, directly tied
to the three distinct states (High/Medium/Low) identified for each site. For instance, with just two sites,
there are nine unique combinations. The number of combinations expands to 27 for three sites, 81 for
four sites, and 243 for five sites. The pattern continues similarly for additional sites. Take the
combinations of four sites as an example, following the copula theory, $P(X^1 < x^1, X^2 < x^2) =$
$f(u^1, u^2) = C(u^1, u^2)$ and $P(X > x) = 1 - P(X < x)$ , the probability formulas of synchronization
are derived as below.
(1) The probability of synchronized high-water is as follows:

$$
\begin{aligned}
P\big(X^1 > X_{ph}^1, X^2 > X_{ph}^2, X^3 > X_{ph}^3, X^4 > X_{ph}^4\big) = 1 - u_{ph}^1 - u_{ph}^2 - u_{ph}^3 - u_{ph}^4 \\
+ C\big(u_{ph}^1, u_{ph}^2\big) + C\big(u_{ph}^1, u_{ph}^3\big) + C\big(u_{ph}^1, u_{ph}^4\big) + C\big(u_{ph}^2, u_{ph}^3\big) + C\big(u_{ph}^2, u_{ph}^4\big) \\
+ C\big(u_{ph}^3, u_{ph}^4\big) - C\big(u_{ph}^1, u_{ph}^2, u_{ph}^3\big) - C\big(u_{ph}^1, u_{ph}^2, u_{ph}^4\big) - C\big(u_{ph}^1, u_{ph}^3, u_{ph}^4\big) \\
- C\big(u_{ph}^2, u_{ph}^3, u_{ph}^4\big) + C\big(u_{ph}^1, u_{ph}^2, u_{ph}^3, u_{ph}^4\big)
\end{aligned}
\tag{6}
$$


(2) The probability of synchronized medium-water is as follows:



$$P = \left(X^1_{pl} < X^1 < X^1_{ph}, X^2_{pl} < X^2 < X^2_{ph}, X^3_{pl} < X^3 < X^3_{ph}, X^4_{pl} < X^4 < X^4_{ph}\right)$$
$$= C\left(u^1_{ph}, u^2_{ph}, u^3_{ph}, u^4_{ph}\right) - C\left(u^1_{ph}, u^2_{ph}, u^3_{ph}, u^4_{pl}\right) - C\left(u^1_{ph}, u^2_{ph}, u^3_{pl}, u^4_{ph}\right)$$
$$-C\left(u^1_{ph}, u^2_{pl}, u^3_{ph}, u^4_{ph}\right) - C\left(u^1_{pl}, u^2_{ph}, u^3_{ph}, u^4_{ph}\right) + C\left(u^1_{ph}, u^2_{ph}, u^3_{pl}, u^4_{pl}\right)$$
$$+C\left(u^1_{ph}, u^2_{pl}, u^3_{ph}, u^4_{pl}\right) + C\left(u^1_{pl}, u^2_{ph}, u^3_{ph}, u^4_{pl}\right) + C\left(u^1_{ph}, u^2_{pl}, u^3_{pl}, u^4_{ph}\right)$$
$$+C\left(u^1_{pl}, u^2_{ph}, u^3_{pl}, u^4_{ph}\right) + C\left(u^1_{pl}, u^2_{pl}, u^3_{ph}, u^4_{ph}\right) - C\left(u^1_{ph}, u^2_{pl}, u^3_{pl}, u^4_{pl}\right)$$
$$-C\left(u^1_{pl}, u^2_{ph}, u^3_{pl}, u^4_{pl}\right) - C\left(u^1_{pl}, u^2_{pl}, u^3_{ph}, u^4_{pl}\right) - C\left(u^1_{pl}, u^2_{pl}, u^3_{pl}, u^4_{ph}\right)$$
$$+C\left(u^1_{pl}, u^2_{pl}, u^3_{pl}, u^4_{pl}\right)$$
$$\tag{7}$$

(3) The probability of synchronized low-water is as follows:

$$P\left(X^1 < X^1_{pl}, X^2 < X^2_{pl}, X^3 < X^3_{pl}, X^4 < X^4_{pl}\right) = C\left(u^1_{pl}, u^2_{pl}, u^3_{pl}, u^4_{pl}\right) \tag{8}$$

**2.3  Stochastic simulation based on RDV-Copula functions**

**2.3.1 Reduced-dimension vine copula construction approach (RDV-Copula) for multi-variate**

To construct joint distribution functions for multiple variables that encapsulate both temporal and spatial relationships, it is essential to incorporate a comprehensive range of information to efficiently capture the interconnections among variables.

Using the flow at $N$ points within a catchment as an example, the relationships among the flows are analyzed. Given that these points reside within the same geographical region, it's highly likely that they are spatially related and the strength of the relationship is negatively correlated with spatial distance. Additionally, each site exhibits temporal correlations, such as the relationship between today's flow and that of the previous day(s), although for simplicity, this analysis assumes relevance only between consecutive days' flows. Incorporating both temporal and spatial dimensions into the analysis implies that for "$N$" sites, there should ideally be "$N + N$" variables considered in constructing the copula function. As the number of sites grows, it simultaneously elevates the dimensionality of the copula, leading to increasingly complex structures. This complexity not only escalates computational efforts but also presents significant challenges in accurately fitting the model. To address this issue, our study introduces a novel methodology termed the Reduced-Dimension Vine Copula Construction Approach (RDV-Copula). This strategy aims to distill essential spatial-temporal information, thereby reducing the vine copula function's dimensionality to simplify the model structure.

The primary goal of this approach is to pinpoint the crucial variables necessary for effectively and efficiently representing the spatial-temporal relationships among different sites. The process begins by identifying variables to capture spatial relationships, under the assumption that the spatial relationships



remain stable over short periods. Consequently, the current day's flows across all sites are selected as
spatial variables, totaling $N$. Subsequently, the Kendall correlation coefficient between the current and
previous day's flows is computed for each site, with the values ranked in descending order. The site with
the highest Kendall coefficient is deemed the most temporally correlated, and its previous day's flow is
also chosen as a key variable for the vine copula construction. Flows from the previous day at other sites
are excluded from being key variables. Ultimately, this approach selects "$N+1$" key variables,
achieving an effective representation of spatial-temporal relationships while minimizing variable count.
The schematic diagram of the process is shown in Figure 3.

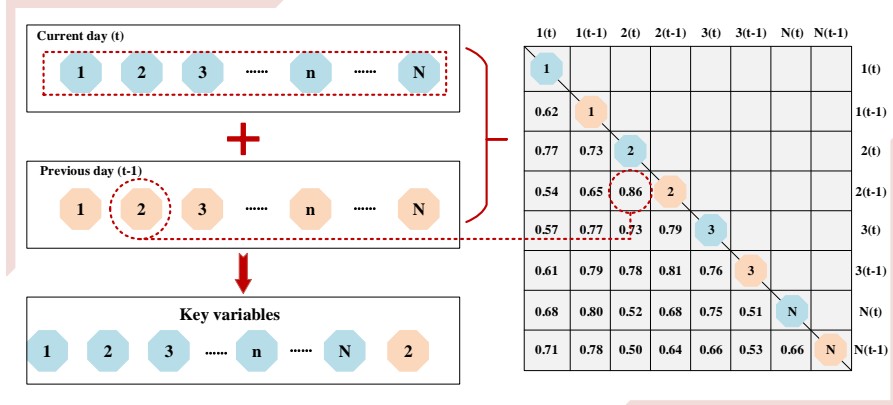

**Figure 3. Schematic diagram of the RDV-Copula method**
**2.3.2 Stochastic simulation**
Simulation methods for multivariate stochastic processes are categorized into two main types:
unconditional and conditional simulations, as delineated by Wu et al. (2015). The core distinction
between these two simulation methods hinges on whether certain data points are pre-determined at the
time of simulation. Figure 4(a) and (b) illustrate the unconditional simulation and the conditional
simulation, respectively.

Unconditional simulation: This simulation approach generates stochastic samples solely based on

the probability distribution of the dataset, without any prior knowledge of data states. All sample data
are produced simultaneously through stochastic simulation, with each data point being in an unknown
state prior to the simulation.

Conditional simulation: Conversely, conditional simulation operates under the premise that data at



specific locations are already known. These known data points are then used to generate random samples,
with the complete set of samples being produced based on both the probability distribution of the data
and the conditions set by the known variables. This method allows for a tailored simulation that
incorporates pre-existing data insights.

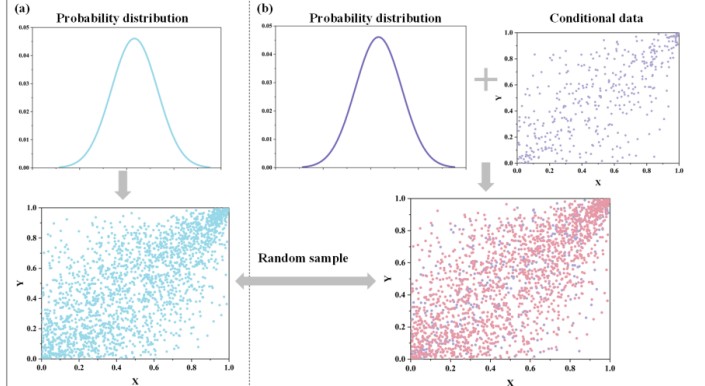


**Figure 4. Schematic diagram for generating random simulation samples (a) unconditional simulation (b)**
**conditional simulation**

Based on the presentation of each section in detail above, it can be generalized that stochastic

simulation based on the RDV-Copula function needs to go through the following steps.

Step 1: Collect as much historical data as possible.

Step 2: Correlation analysis is conducted on the collected data by calculating the Kendall's

coefficient.

Step 3: According to the method of filtering key variables proposed in Subsection 2.3.1, the

representative key variables are extracted based on the correlation relationship among multiple variables.

Step 4: Marginal distribution functions are fitted to the historical data series of the screened key

variables.

Step 5: Based on the proposed RDV-Copula approach, the joint distribution function of multi-site

runoff sequences is constructed with consideration of spatial-temporal relationships.

Step 6: The stochastic simulation sequences of runoff are generated by performing unconditional

stochastic simulation and conditional stochastic simulation based on the constructed vine copula
functions with different structures.



**3 Case study**
**3.1 Study area and data description**
This study applies its methodology to a case study focusing on constructing spatial-temporal
relationships within the Shifeng Creek area, located in the Jiaojiang River catchment in Eastern China.
The Jiaojiang River ranks as the third largest river in Zhejiang Province. As the primary tributary of the
Jiaojiang River basin and the principal watercourse in Tiantai County, Shifeng Creek plays a significant
role. Rainfall distribution in the Shifeng Creek catchment is notably uneven throughout the year, with a
substantial portion, approximately 70 to 80%, occurring from March to September. The remaining 20 to
30% of yearly rainfall is distributed over the other months. The period from July to September is
particularly marked by intense storms and rainfall, largely influenced by the Pacific subtropical high-
pressure system and the frequent occurrence of typhoons, contributing about 35% of the annual total
precipitation, with amounts ranging from 400 to 600mm.
The objective of this study is to delineate the spatial-temporal relationships of inflows within the
catchment during August, a flood-prone month, to enhance flood pattern understanding and support
effective flood management strategies. In the Shifeng Creek region, there are many important hydraulic
structures and critical control cross-sections. This study focuses on four major sites within the Shifeng
Creek catchment: the Lishimen Reservoir (LSM) site, the Longxi Reservoir (LX) site, along with the
Qianshan (QS) cross-section site and the Shaduan (SD) cross-section site. These sites are strategically
located along the upper, middle, and lower stretches of Shifeng Creek, facilitating a comprehensive
analysis of the entire catchment and flood characteristics of Shifeng Creek. To achieve this, daily runoff
data of August, covering a span from 2000 to 2020, have been compiled. This dataset encompasses
inflows for the LSM and LX reservoir sites, as well as flow data for the QS and SD cross-sections. The
geographic positioning of Shifeng Creek is depicted in Figure 5.





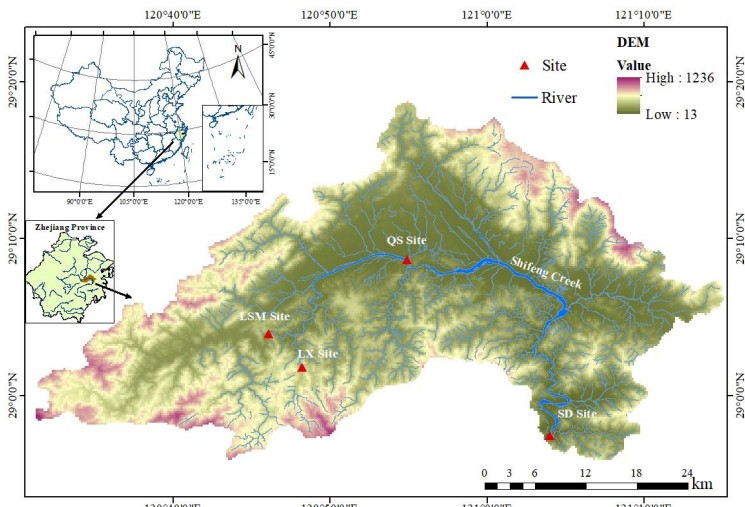


**Figure 5. Map of location of Shifeng Creek**

**3.2   Numerical experiments setup**

**3.2.1 Synchronization frequency analysis based on spatial relationship**

In this study, we employ the vine copula function to construct the joint distribution of runoff across four

sites, aiming to analyze the synchronization frequency of floods in August, a month identified as having

a high risk of flooding. The variables under consideration include the inflow from these four sites,

denoted as LSM-Aug, LX-Aug, QS-Aug, and SD-Aug. Our initial step involves calculating the Kendall

coefficients among these variables to assess their interdependencies. Following the methodology outlined

in Subsection 2.1.1, we determine the marginal distribution functions of the four variables through a

fitting test. Subsequently, based on the marginal distribution function of each variable, the joint

distribution function of four variables is constructed. The parameters of the vine copula are estimated via

the maximum likelihood method, with the Akaike Information Criterion (AIC) serving as the selection

criterion to ensure optimal model fit. Upon passing the fitting test, we identify the most appropriate vine

copula structure to accurately model the relationships among the variables.

With the four-dimensional vine copula function established, we proceed to calculate and analyze

the synchronization frequency of inflows as described in Subsection 2.2. The inflows at the four sites are

symbolized as LSM, LX, QS, and SD, with high-water and low-water inflow amounts represented as

$X_{ph}$, $Y_{ph}$, $Z_{ph}$, $W_{ph}$ and $X_{pl}$, $Y_{pl}$, $Z_{pl}$ and $W_{pl}$, respectively. The marginal distribution functions are





denoted as $u$, $v$, $r$   and $s$.

Considering the three potential states (High/Medium/Low) at each site, a total of 81 possible inflow-

state combinations are identified. Among these, the combinations [X-H, Y-H, Z-H, W-H], [X-M, Y-M,
Z-M, W-M], and [X-L, Y-L, Z-L, W-L] are classified as synchronous, while the remainder are deemed
asynchronous. The calculation equations can be referenced in Appendix B.
**3.2.2 Various vine copulas construction based on spatial-temporal relationships and stochastic**
**simulation**
To enhance the vine copula function's accuracy, it's imperative to integrate the temporal dimension into
its construction. In this section, the vine copula functions are developed on a daily basis, encompassing
a series of 31 copula models corresponding to each day of August, from the 1st to the 31st. Consequently,
both Kendall correlation analysis and the fitting of marginal distribution functions must be independently
conducted for the data spanning these 31 days. Following this preliminary analysis, 31 distinct
relationship functions are constructed, each tailored to the specific type of vine copula identified for each
day.
**3.2.2.1    RDV-Copula function construction**
Given that all four sites are situated within the Shifeng Creek watershed, their spatial interconnectivity
is inherent and can be leveraged in constructing a vine copula function. Additionally, due to the persisting
effects of rainfall, the flow at any given site is also temporally linked to its previous day's flow. To
encapsulate this temporal correlation, the study integrates the inflows from the four sites over two
consecutive days. The inflows for the current day are denoted as LSM, LX, QS, and SD, while those for
the previous day are labeled LSM1, LX1, QS1, and SD1, respectively.

The methodology, as detailed in Subsection 2.3, initiates by analyzing the current day's inflows at

the four sites to establish their spatial relationships. The subsequent step involves identifying the site
with the most significant correlation to its preceding day's inflow, which is then used as a as a variable
to represent the temporal relationship on that day. For instance, analysis between August 1st and 2nd
reveals that the LSM site had the highest correlation with its prior day's flow compared to the other sites.
Taking the construction of the copula function relationship between August 1st and August 2nd as an
example, the analysis reveals that the LSM site has the highest correlation with its previous day's flow
compared to the other three sites. As a result, a total of five key variables are determined for this





relationship set, including LSM, LX, QS, SD, and LSM1, effectively encompassing both temporal and
spatial correlations while streamlining the variable dimensions within the copula.

Due to the fundamental difference in structure between C-vine and D-vine copula, this study

constructs five-dimensional RDV-Copula functions based on these two types, respectively, labeled as
RDV-Cvine and RDV-Dvine. These two types of models should first be evaluated against each other on
various indexes, including AIC, BIC, and Loglik, to ascertain the most suitable five-dimensional RDV-
Copula structure. This chosen structure is then further compared with other copula functions to validate
its efficacy.
**3.2.2.2     Benchmark copula functions construction**
To validate the effectiveness of the RDV-Copula approach, this study compares it against a series of
benchmark copula functions. These benchmarks are constructed by applying various combinations of
multiple variables and stochastic simulation approaches to the existing data, resulting in vine copula
models of differing dimensions. The specifics of these vine copula models are summarized as follows
and illustrated in Figure 6.

**Benchmark 1:**

Focuses solely on spatial correlations, utilizing inflows at the four sites on the current day (LSM-

LX-QS-SD) to create a four-dimensional vine copula. Simulations are conducted unconditionally.

**Benchmark 2:**

Incorporates both spatial and temporal correlations, including inflows at the four sites for both the

current and previous day (LSM-LX-QS-SD-LSM1-LX1-QS1-SD1), resulting in an eight-dimensional
vine copula. This model also employs unconditional simulation.

**Benchmark 3:**

Like Benchmark 2, this model considers both spatial and temporal correlations using the same set

of key variables (LSM-LX-QS-SD-LSM1-LX1-QS1-SD1), thereby forming an eight-dimensional vine
copula. However, it differs in its application of conditional simulation, assuming the previous day's runoff
as a known condition to simulate the current day's flows.

To further detail the distinctions in stochastic simulation approaches, the RDV-Copula functions are

bifurcated into two categories:

**RDV-un/ RDV-con:**




Both models account for spatial and temporal correlations by incorporating inflows at the four sites
on the current day and the inflow at one site from the previous day (LSM-LX-QS-SD-X1), creating a
five-dimensional vine copula. The variable "X" represents the site with the strongest temporal connection.
The "RDV-un" employs unconditional simulation, while "RDV-con" utilizes conditional simulation.

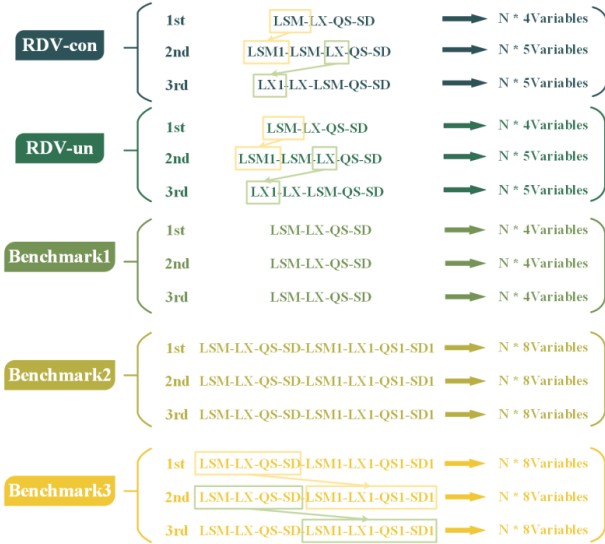


**Figure 6.   Five different vine copula models**
**4    Results**
**4.1   Synchronization frequency analysis**
Prior to performing a synchronization frequency analysis on multiple variables, it is imperative to
conduct a correlation analysis to verify the presence of spatial correlations among them. Following the
approach outlined in Subsection 2.1, this study begins with a correlation analysis of the daily runoff in
August at the four selected sites, utilizing Kendall coefficients to quantify their interconnections. The
results of this analysis, demonstrating the correlation among the four variables, are shown in Figure 7(a).
Subsequent to identifying correlation, the next step involves determining the marginal distributions for
these variables. Figure 7(b) displays the results of this process, showcasing both the plots of the fitted
marginal distributions for the four variables and the actual data distribution, thereby laying the
groundwork for a comprehensive understanding of the data's distribution characteristics.





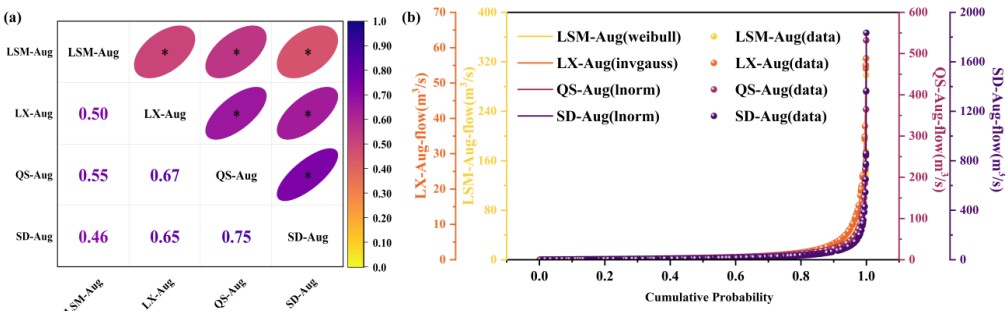

Figure 7. (a) Results of correlation analysis for daily runoff at multiple sites (b) Cumulative probability

distribution of the preferred marginal distribution function

Figure 7 demonstrates that the correlations among the four study variables have all passed the

significance test ($p \leq 0.05$), with the QS and SD sites exhibiting the strongest correlations. This is

closely followed by the spatial connections between the LX site and both QS and SD sites, with

correlation coefficients of 0.67 and 0.65, respectively. The correlations involving the LSM site and the

other three sites are relatively low, reflecting a reduction in spatial correlation with increasing distance.

In terms of runoff distribution, the LSM site's runoff adheres to the Weibull distribution (weibull), while

the runoff at the LX site fits the Inverse Gaussian distribution (invgauss), and the runoffs at both QS and

SD sites align with the Log-normal distribution (lnorm). Building on the vine copula function

methodology outlined in Subsection 2.1.2, we have developed a four-dimensional vine copula function

using these variables. The function's structure, alongside the estimated parameters, is detailed in Table 1.

Table 1 Four-dimensional vine copula structure and parameters

| Tree | edge | family | rotation | parameters | tau | loglik |
|------|------|--------|----------|------------|-----|--------|
| | 1,3 | bb7 | 0 | 2.2, 1.1 | 0.54 | 296 |
| 1 | 2,3 | t | 0 | 0.86, 6.51 | 0.66 | 433 |
| | 3,4 | t | 0 | 0.92,2.69 | 0.74 | 636 |
| | 1,4\|3 | frank | 0 | -1.3 | -0.15 | 15 |
| 2 | 2,4\|3 | Bb1 | 180 | 0.13, 1.10 | 0.15 | 25 |
| 3 | 12\|43 | bb7 | 180 | 1.07, 0.21 | 0.13 | 24 |

Upon the construction of four-dimensional vine copula function, the synchronization frequency

analysis can be expanded. Using the approach detailed in Subsection 2.2, we obtained 81 encounter



probabilities reflecting potential inflow scenarios at four sites: high-water, medium-water, and low-water.
Figure 8(a) shows these 81 probabilities in detail. Figures 8(b)-(g) present aggregated views, focusing
on nine combinations representing two of the four variables in each of their three states.

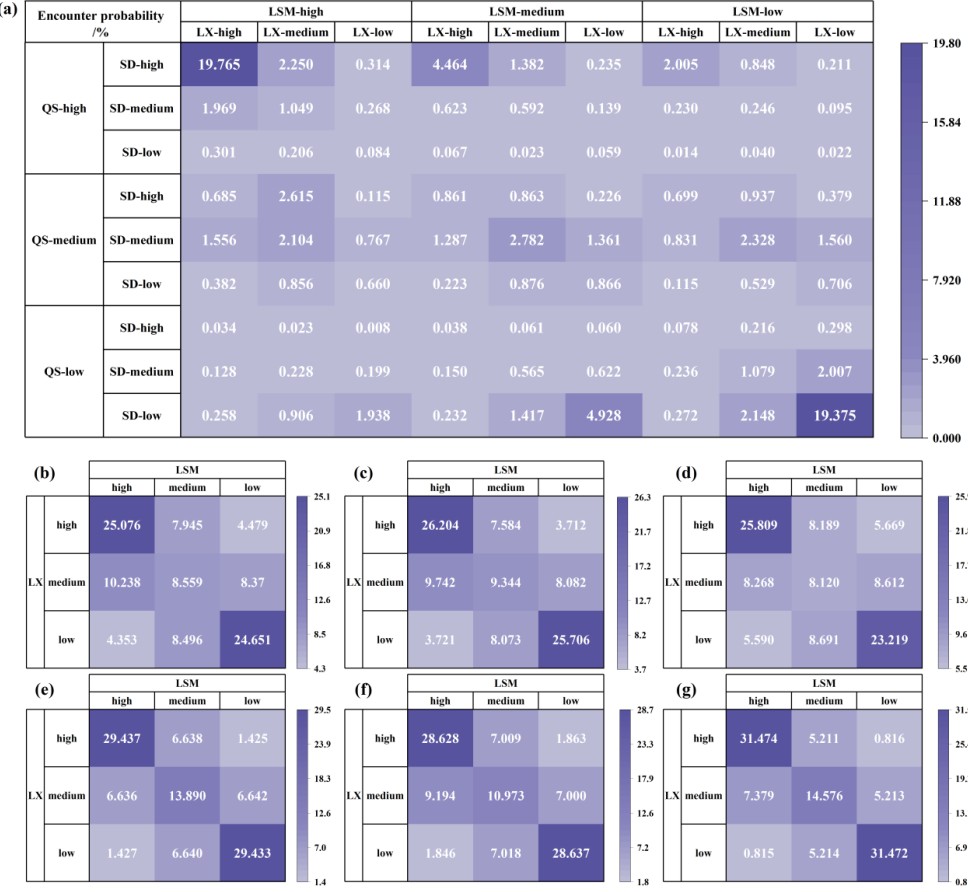

**Figure 8.   Encounter probabilities for the multiple sites (a) LSM-LX-QS-SD (b) LSM-LX (c)**
**LSM-SD (e) LS-QS (f) LX-SD (g) QS-SD**
As observed in Figure 8, the cumulative probability of synchronization across all four sites
simultaneously stands at 41.92%, encompassing three scenarios: (1) LSM-high, LX-high, QS-high, SD-
high (2) LSM-medium, LX-medium, QS-medium, SD-medium (3) LSM-low, LX-low, QS-low, SD-low.
Any two of these sites also demonstrate a very strong synchronization between them, with probabilities
nearing 60%. The obvious dark colored blocks in the graph indicate the high probabilities of being the
high-water or the low-water concurrently. Among these, the strongest synchronization occurs between



the QS and SD sites, reaching a probability of 77.52%. This is closely followed by the LX site's
synchronization with both QS and SD sites, at probabilities of 72.76% and 68.24%, respectively. While
the LSM site's synchronization probabilities with the other sites are comparatively lower, they still exceed
50%, recorded at 58.29% with the LX site, 61.25% with the QS site, and 57.15% with the SD site. This
analysis underscores the clear spatial correlation among the four sites and highlights the critical
importance of monitoring high-water synchronization. This is because such a case of simultaneous high
water at multiple sites can easily induce flooding and pose a risk to the downstream. By analyzing the
relationship of flow among multiple sites in advance and clarifying the probability of synchronization, it
would be more conducive to the formulation of flood control and scheduling strategies to reduce the
probability of flood encounters and protect the safety of the downstream.
**4.2  Construction of joint distributions of multi-site daily inflows**
**4.2.1 Correlation analysis**
Correlation analysis serves as an efficient tool for quickly identifying and quantifying the correlations
among multiple variables. Following the methodology outlined in Subsection 2.1, this study incorporates
both temporal and spatial correlations in its analysis. To achieve this, historical runoff data from four key
sites, along with the previous day's runoff data at each site, were used, resulting in a set of eight variables
for the correlation analysis. The results of the analysis are presented in Figure 9. Due to the large amount
of information, only part of the correlation results is shown here. The complete set of results is available
in Appendix C.

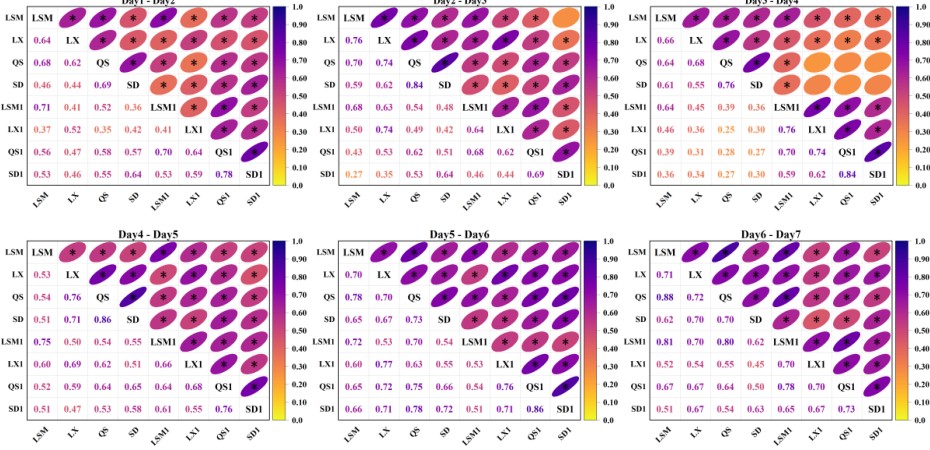



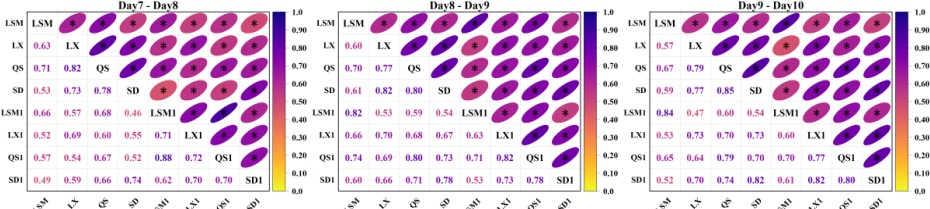

**Figure 9. Partial results of correlation analysis for daily runoff at multiple sites (LSM, LX, QS, SD**
**represent the runoff sequences of current day, while LSM1, LX1, QS1, SD1 represent the runoff sequences**
**of previous day)**
Figure 9 illustrates the Kendall correlation coefficients between pairs of variables. The intensity of
colors correlates with the strength of positive correlation, with darker shades signifying a correlation
coefficient closer to 1. The "*" on the ellipse means that the correlation passes the significance test of
$\alpha = 0.05$. This figure uncovers a marked positive correlation among the runoff series at the LSM, LX,
QS, and SD sites, with approximately 93% of these correlations meeting the significance threshold. This
finding indicates that there is an obvious spatial correlation among the four locations. Notably, the QS
and SD sites exhibit the strongest spatial correlation, with an average coefficient in August of 0.74,
closely followed by the LX reservoir's correlation with the QS and SD sections at 0.67 and 0.63,
respectively. In comparison, the LSM reservoir's runoff shows relatively lower correlations with the other
sites, averaging 0.48 with LX site, 0.55 with QS site, and 0.45 with SD site in August.
Upon analyzing the temporal correlation of runoff at each site for adjacent days within August
(denoted as LSM-LSM1, LX-LX1, QS-QS1, SD-SD1), it becomes evident that temporal correlations are
significant and should not be overlooked. Particularly in early August, these correlations register at a
notably high level, suggesting more frequent flooding during this period. The LSM site demonstrates a
standout temporal correlation, averaging 0.72 in August, indicative of a strong link between the current
and previous day's runoff. The other sites display slightly lower, yet significant, temporal correlations:
LX at 0.65, QS at 0.65, and SD at 0.67. When these temporal correlations are considered alongside the
spatial ones, it's evident that LSM's temporal correlation surpasses its spatial correlation with other sites.
These correlation analysis results solidly confirm both spatial and temporal correlations among the
four sites, laying a foundational basis for advancing with the construction of a copula structural model.



**4.2.2 Fitting of marginal distribution of each runoff**

In this study, twelve distinct distribution functions were utilized to model the daily runoff at four sites throughout August. To assess the goodness-of-fit of these distributions, the Kolmogorov-Smirnov (K-S) test, with a significance level of 0.05, was employed. Following a successful significance test, the Akaike Information Criterion (AIC) minimum method was applied to evaluate and determine the optimal marginal distribution for each dataset. Figure 10 shows the preferred marginal distribution functions for each variable over the 31 days of August. This figure contrasts the actual historical data points against the curves of the fitted functions, offering a visual representation of the fitting accuracy. The specific marginal distribution functions chosen for each variable, along with their parameters for each day, are comprehensively listed in Appendix D. Figure 10 notably illustrates how well these selected marginal distribution functions match the actual data for all four variables from the 1st to the 12th of August. The chosen marginal distribution functions for the entire month are detailed in Figure D1. Furthermore, the figure's legend explicitly details the types of fitting functions employed for each variable, providing a clear and comprehensive overview of the distributional characteristics.

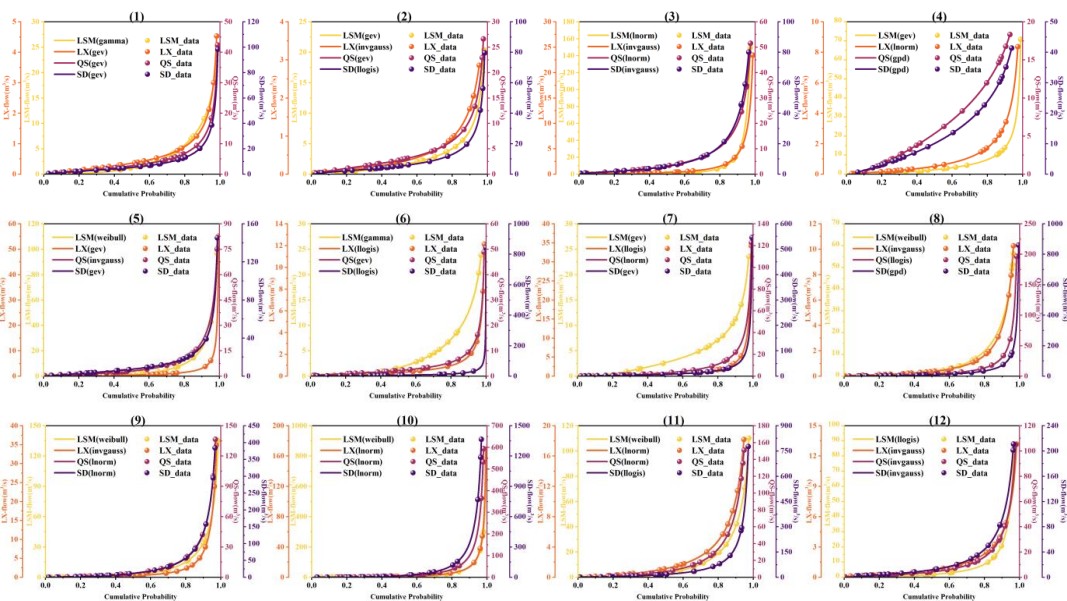

**Figure 10. Cumulative probability distribution of the preferred marginal distribution function for runoff on each day throughout 1st-12th in August**

The distribution of the corresponding marginal distribution functions for the four variables over the



31 days in August is summarized in Figure 11.

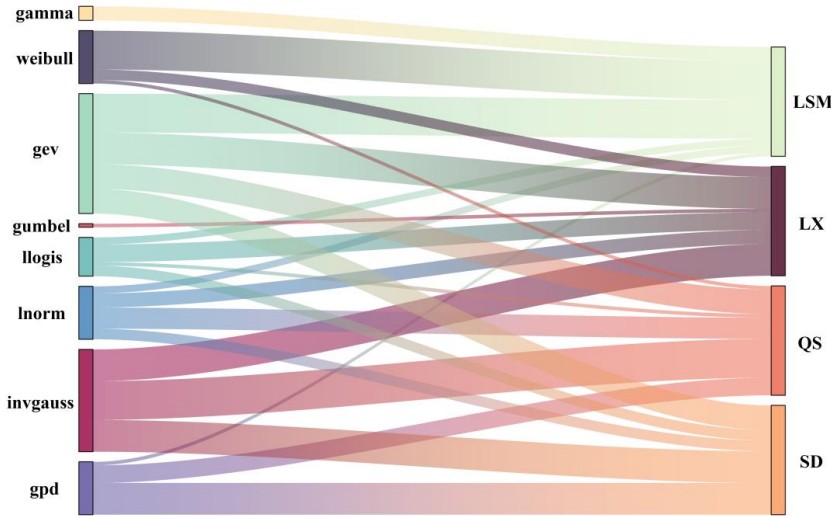


**Figure 11.**    **Distribution of the preferred marginal distribution function for the daily series of flows at LSM,**
**LX, QS and SD site in August**
Figure 11 shows that most streamflow series follow the "gev" distribution (27.52%) and the
"invgauss" distribution (23.39%). Relatively few streamflow series follow the "weibull", "llogis",
"lnorm", and "gpd" distributions, and only a very small number follow the "gamma" and "gumbel"
distributions. Additionally, 71% of the runoff sequences at the LSM site follow the "weibull" and "gev"
distributions, each accounting for 35.5%. The runoff sequences at the LX site, the QS site, and the SD
site predominantly follow the "gev" and "invgauss" distributions, accounting for 29.03% and 29.03% at
the LX site, 22.58% and 35.48% at the QS site, and 22.58% and 29.03% at the SD site, respectively.
Meanwhile, nearly 30% of the runoff sequences at the SD site also follow the "gpd" distribution.
**4.2.3 Construction of RDV-Copula function**
Following the identification of each variable's marginal distribution, the next step involves selecting the
appropriate copula structures to construct the vine copula models among the multiple variables. Utilizing
the RDV-Copula function construction approach described in Section 3.2.2.1, we identified the sites
exhibiting the highest temporal correlation for each day in August, based on our correlation analysis
results. The variables chosen for each specific day are illustrated in Figure 12.



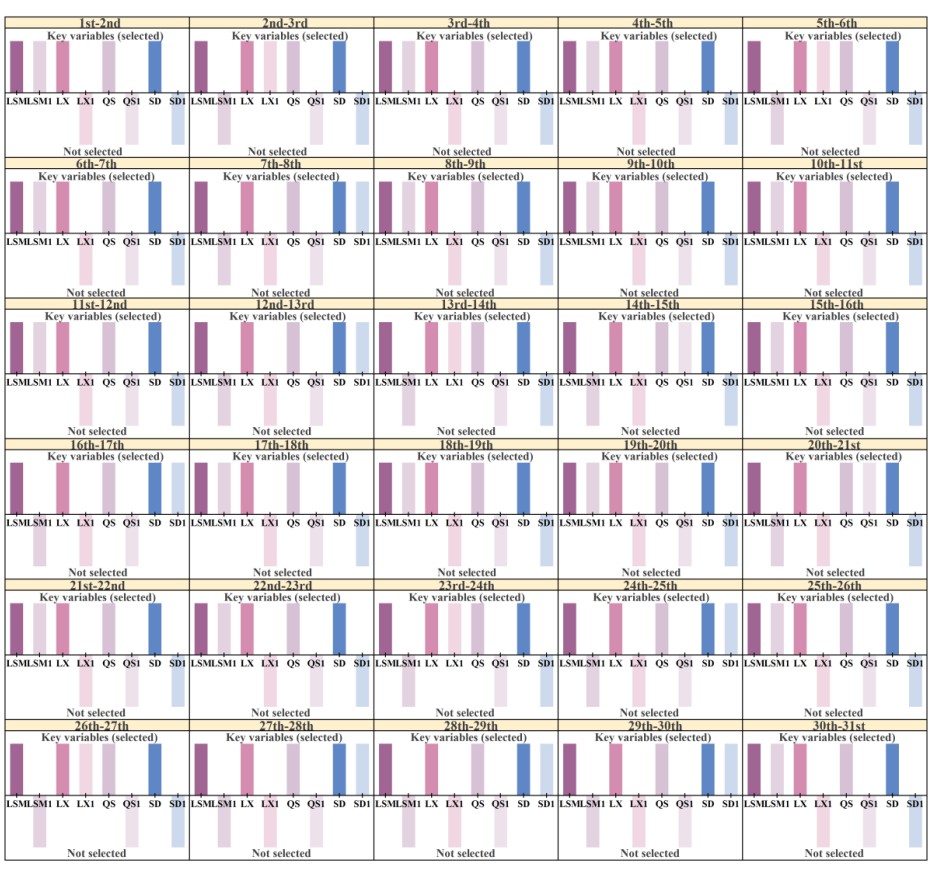

**Figure 12. Key factors in the five-dimensional vine copula structure constructed in two adjacent days**

**(LSM, LX, QS, SD represent the runoff sequences of current day, while LSM1, LX1, QS1, SD1 represent the**

**runoff sequences of previous day)**

Prior to selecting a specific copula function for modeling, it is essential to decide on the type of

copula to be employed. Among the options, C-vine and D-vine structures stand out for their common use

in various applications. In this study, we constructed both C-vine and D-vine copula structures for the set

of multiple variables under consideration. To evaluate the efficacy of these structures, metrics such as

the Akaike Information Criterion (AIC), Bayesian Information Criterion (BIC), and Log-Likelihood

(Loglik) values were utilized and computed, with the results presented in Figure 13. The AIC and BIC

values reveal that, for the majority of cases, the D-vine copula structures exhibit significantly lower

values compared to those of the C-vine structures. Lower values in these criteria suggest a model's better



performance and fit. Moreover, the comparison of log-likelihood values also showed that D-vine
structures typically yielded lower values than their C-vine counterparts. Consequently, the D-vine copula
structure was identified as more effective and suitable for modeling the intricate relationships among the
variables in this study. Therefore, the RDV-Copula and other benchmark copula models were designed
using the D-vine structure.

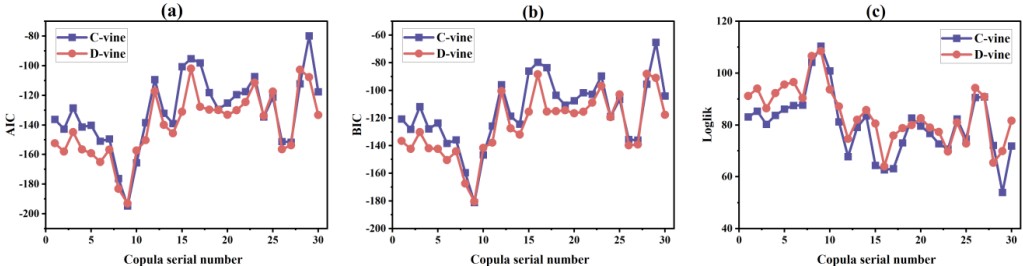

**Figure 13.   Comparison of the performance of RDV-Copula models for C-vine and D-vine (a) AIC (b) BIC**
**(c) Loglik**

A large number of copula families were utilized to model the joint distributions, such as Gaussian

copula, Gumbel copula, t copula and so on. Following the guidance of AIC criteria, the most suitable
pair-copula for each connection within every tree was selected. After fitting the goodness of the copula
functions, we employed the maximum likelihood method to estimate the parameters. As an illustrative
example, the copula structure for August 1st-2nd is shown in Figure 14. This figure not only reveals the
best-fit copula family for each pair of adjacent nodes but also the estimated parameters. The nodes,
labeled 1 through 5, represent LSM, LX, QS, SD, and X1, which indicates the site with the highest
temporal correlation on that day, respectively. In this instance, X1 corresponds to LSM1. It is important
to note that the specific choice of X1 might vary from day to day, as further elaborated in Figure 12. In
Figure 14, each pair of subfigures situated between nodes shows two aspects of the bi-dimensional copula
function for those nodes. The first subfigure presents the joint probability plot, while the second
illustrates the joint probability density plot.





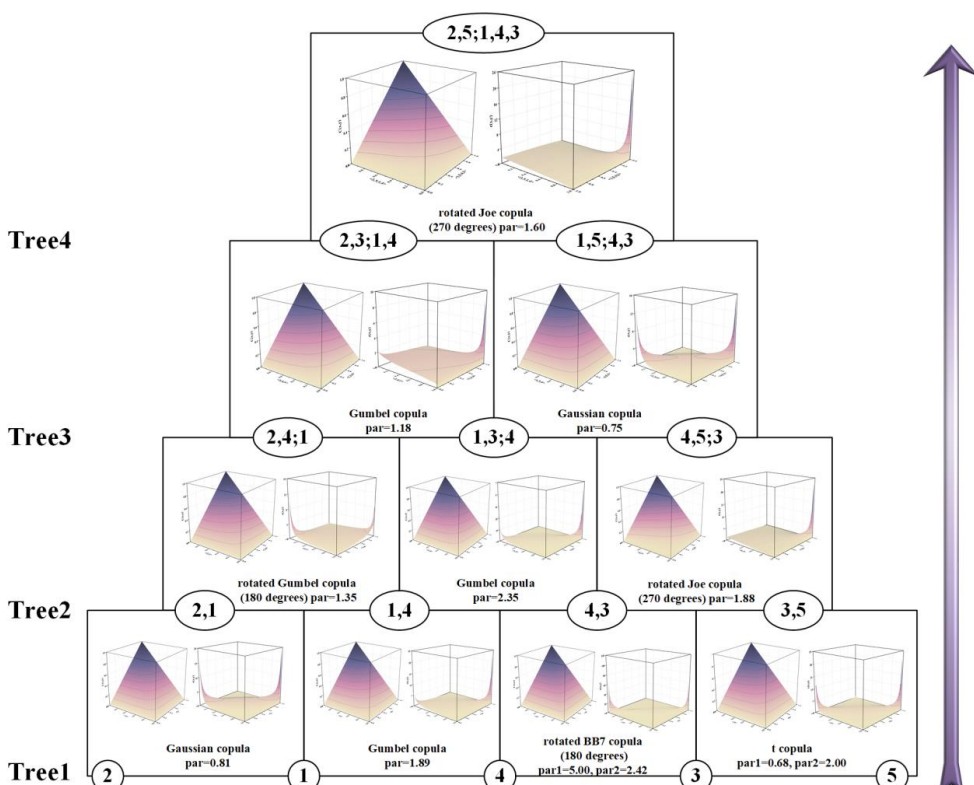

**Figure 14. Structure of the five-dimensional D-vine copula model for August 1st -2nd (Nodes 1–5 represent LSM, LX, QS, SD, and LSM1; The plots between each two nodes are schematic plots of the corresponding copula function, with joint probability plot on the left and joint probability density plot on the right.)**

## 4.3 Stochastic simulation results of runoff from multiple sites

To validate the models and facilitate a comparative analysis of different vine copula functions, the following work was carried out. Initially, the constructed copula structure and the results from parameter estimation were incorporated into a simulation process, generating 20,000 sets of random runoff scenarios for each day in August. Considering August's susceptibility to flooding and the typical continuity of rainfall events, it's highly likely that runoff on consecutive days is temporally correlated. Therefore, comparing only the mean and standard deviation of runoff simulated for individual days might not fully capture the model's simulation efficacy. In this context, the study calculated the mean and standard deviation for the current day by considering the simulated flows of both the preceding and following days. Ultimately, after the exclusion of outliers from the 20,000 sets of simulated runoff



scenarios, the average of the mean and standard deviation calculated from these three days' simulated
flows will be used as the mean and standard deviation for the current day. The runoff simulation results
for the four locations (LSM, LX, QS, and SD) are presented in Figures 15, 16, 17 and 18, respectively.
Notably, in each figure, subfigure (a) displays the mean values and standard deviations from the
simulation results for the five copula structures, allowing these results to be compared against historical
observations for a nuanced evaluation of the simulation's performance. Subfigures(b), (c), (d), (e) and (f)
represent the simulation results for five different sets of copula structures (RDV-con, RDV-un,
Benchmark1, Benchmark2 and Benchmark3) respectively. The solid line in the figure is the mean of the
simulation results and the shaded area represents the uncertainty (±1 standard deviation) of the simulation.

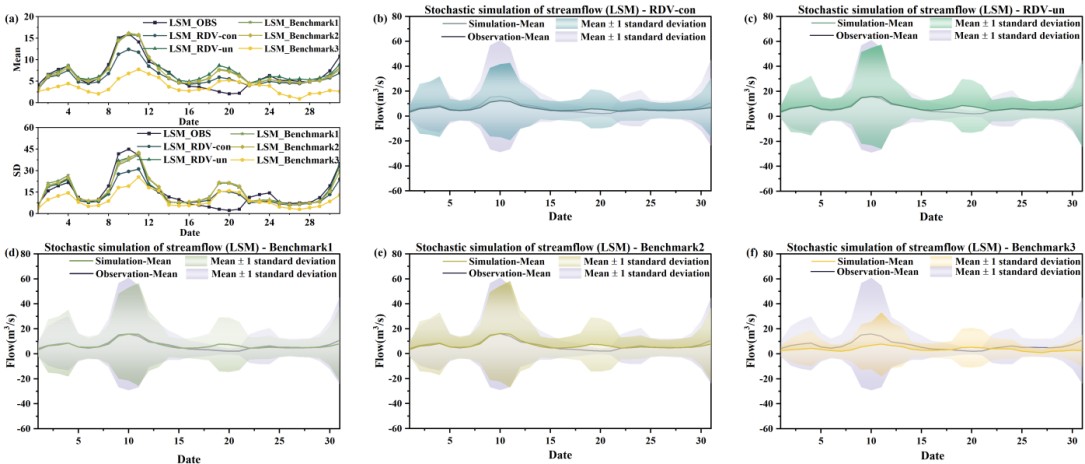

**Figure 15.   Comparison of the actual observed series with simulation results of four copula structures at**
**LSM site (a) comparison of daily runoff mean values and standard deviation (b) simulation results of RDV-**
**con (c) simulation results of RDV-un (d) simulation results of Benchmark1 (e) simulation results of**
**Benchmark2 (f) simulation results of Benchmark3**




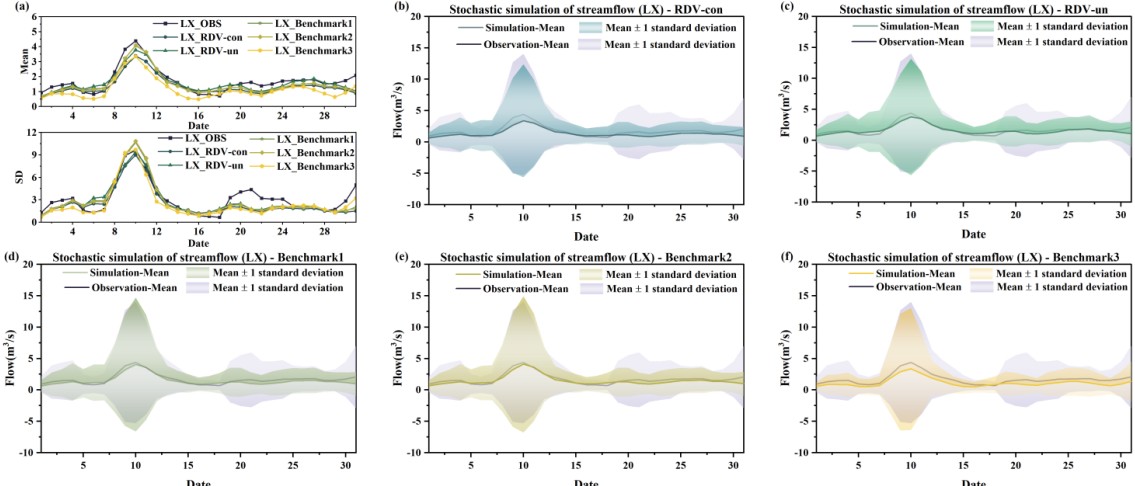

**Figure 16. Comparison of the actual observed series with simulation results of four copula structures at LX site (a) comparison of daily runoff mean values and standard deviation (b) simulation results of RDV-con (c) simulation results of RDV-un (d) simulation results of Benchmark1 (e) simulation results of Benchmark2 (f) simulation results of Benchmark3**

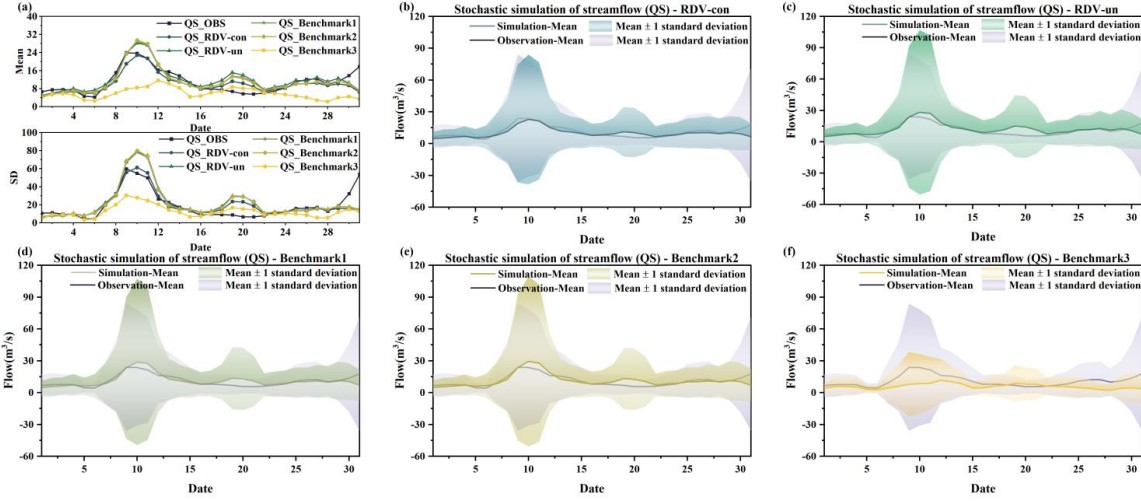

**Figure 17.   Comparison of the actual observed series with simulation results of four copula structures at QS site (a) comparison of daily runoff mean values and standard deviation (b) simulation results of RDV-con (c) simulation results of RDV-un (d) simulation results of Benchmark1 (e) simulation results of Benchmark2 (f) simulation results of Benchmark3**





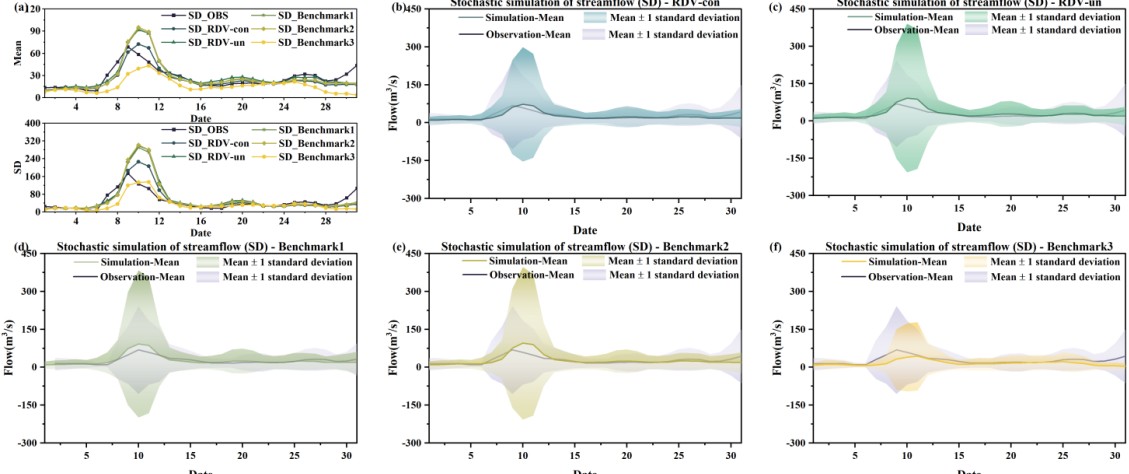

**Figure 18. Comparison of the actual observed series with simulation results of four copula structures at SD site (a) comparison of daily runoff mean values and standard deviation (b) simulation results of RDV-con (c) simulation results of RDV-un (d) simulation results of Benchmark1 (e) simulation results of Benchmark2 (f) simulation results of Benchmark3**

From four figures, it is evident that the simulation results of RDV-Copula, Benchmark1 and Benchmark2 are comparatively more accurate. The mean values and standard deviations from these simulations closely match the actual observed runoff, particularly for simulations involving smaller flow magnitudes, where the accuracy aligns more precisely with the actual values. Although the RDV-Copula results are consistent with the benchmark models, they do not exhibit a marked advantage for smaller flows. However, in scenarios involving larger flows, such as those at the SD site, RDV-Copulas outperform other models, highlighting their superiority in capturing the characteristics of larger inflow events. This analysis suggests that for smaller flows, models focusing solely on spatial relationships suffice to capture the critical interrelationships among variables. In contrast, for larger flows, neglecting the influence of temporal correlations can lead to substantial inaccuracies in the simulation results, suggesting that larger flows are more significantly influenced by adjacent day's flows. Comparing the four figures, we can also find that the simulation results at LX location consistently exhibit high accuracy, with the simulation results basically covering the actual observations. This suggests that the constructed copula models can easily extract the historical correlations and simulate them, particularly in smaller flow magnitudes.



However, the Benchmark3 model's performance is notably less effective among the five models.
This suboptimal performance can be attributed to two main factors. Firstly, the complexity of the eight-
dimensional copula function, which involves a diverse combination of "trees," "nodes," and various types
of parameters, poses significant challenges in accurately extracting the relationship characteristics among
the four sites. Secondly, the conditional simulation approach of Benchmark3, which relies on the previous
day's flow at the four sites as a known condition for simulation, is highly susceptible to the accuracy of
these initial conditions. If the simulation results for the previous day contain significant errors, these
inaccuracies are likely to propagate through the simulation, leading to compounded errors in the entire
results. Another noteworthy point is that the simulation results on the August 10th, 20th and 31st are not
quite consistent with historical conditions. This is because the runoff on these three days has been at a
low level for most of the time over a number of years in history. It is therefore a rather exceptional
phenomenon that a major flood event occurred on these particular dates in just one year. Specifically, the
data recorded on these dates (August 10, 2009, August 31, 2011, and August 20, 2014) indicate unusually
high runoff, which significantly exceeds their respective historical averages. Such an occurrence presents
a challenge for the simulations, as it requires accurately capturing and replicating these atypically high
flow values within the model.
Comparing the two types of simulations of RDV-Copula, it can be found that the performances of
the simulation results of RDV-un and RDV-con are similarly well for LSM and LX sites. However, in
the simulation of QS and SD sites, RDV-con shows an obvious superiority compared to RDV-un. This
illustrates the better generalization of conditional simulation for such complex structure with spatial-
temporal relationships. In contrast to the unconditional simulation, RDV-con can better utilize the
temporal correlation to improve the accuracy of the simulation. Meanwhile, since it is different from the
conditional simulation of the eight-dimensional vine copula (Benchmark2), RDV-con successfully
reduces the cumulative error caused by the excessive dimensionality.
In summary, for the relational construction and stochastic simulation of flows across varying
magnitudes, RDV-Copula and Benchmark2 emerge as more suitable, particularly when considering the
influences of both temporal and spatial correlations. However, the use of an eight-dimensional copula
function in Benchmark2 introduces significant computational demands and adds complexity to the
problem. RDV-Copula is favored for its effective integration of temporal and spatial correlations, while



also simplifying the copula structure, thereby streamlining the problem-solving process and enhancing
computational efficiency.

## 5    Discussion

For variables with interdependencies, the copula function, increasingly popular in contemporary studies,
extracts spatial-temporal relationships from their marginal distributions. Vine copulas are notably
effective in modeling complex dependencies among variables, as they offer substantial flexibility. This
capability is exemplified in the work of Pereira and Veiga (2018), who developed a multivariate
conditional model using D-vine copulas for simulating periodic streamflow scenarios, emphasizing the
structured arrangement of variables to capture monthly flow dependencies. This and numerous other
studies (Nazeri Tahroudi et al., 2022; Wang et al., 2018, 2019; Wang and Shen, 2023a) underscored the
effectiveness of vine copulas in capturing dependencies among variables with differing marginal
distributions.

The synchronous probability analysis of multi-site runoff shows that the vine copula model can be

used to provide a good fit to the dependencies among variables obeying different marginal distributions.
Similar conclusions have been obtained in other studies (Qian et al., 2022; Ren et al., 2020; Wei et al.,
2023). In the study of Xu et al. (2022), the multivariate Copula model was implemented to evaluate the
synchronous–asynchronous characteristics for hydrological probabilities for the multiple water sources.
The simultaneous probabilistic analysis of multi-site runoff provides an understanding of the flood
characteristics of the catchment leading to better flood control and prevention.

For high-dimensional variable dependency analysis, the structure of the vine copula is extremely

complicated to construct. Depending on the number of hydrometric stations, Wang and Shen (2023b)
established the 7-dimensional regular vine (R-vine) copula models to depict the complex and diverse
dependence. To tackle the problem above, in their study, the corresponding vine structure was specified
by the vine structure array that can reflect the sequence of tributaries flowing into the main stream and
the spatial locations of different hydrometric stations. The performance of the ultimate simulation results
was favorable, but it did not incorporate the temporal connection of the variables for each hydrometric
station. If considered, it would lead to an exponential increase in the dimensionality of the variable. The
RDV-Copula method proposed in this study aims to minimize the dimensionality of the copula model



while extracting the effective information of spatial-temporal relationships. The evaluation criterion of
high-performance stochastic simulation is that the simulated series can preserve the statistical
characteristics of the observed records (Hao and Singh, 2013). As shown in Figure 15 - 18, different vine
copula structures have a large impact on the results of stochastic simulations. The simulation results of
the four-dimensional and five-dimensional vine copula models are relatively closer to the actual historical
values. Although the eight-dimensional vine copula model takes more variables into account, including
both temporal and spatial correlation, the model is too complicated due to many variables, which makes
the simulation less efficient on the contrary. This illustrates that when performing multi-site runoff
simulations, it is not better for the vine copula function to consider as many variables as possible.
Compared to the four-dimensional copula structure that only considers spatial relations, the five-
dimensional copula structure can better fit the characteristics of high flows, which is especially evident
in the simulation results of QS and SD points. This is due to the fact that high flows in flood season
mostly originate from continuous heavy rainfall, which implies that the temporal connection is not
negligible for capturing the flow characteristics.
Consequently, the approach introduced in this study effectively integrates all pertinent information
for multi-site runoff simulations while reducing the complexity of the vine copula function. This
methodology strikes a critical balance between detailed representation and practicality in model
complexity, enhancing the applicability of the simulations.
**6   Conclusions**
This study introduced an innovative approach designed to capture the spatial-temporal relationships
across multiple sites while simplifying the computational complexity inherent in vine copula functions.
By computing Kendall correlation coefficients, we assessed the interconnections among various sites.
Utilizing the approach proposed, we pinpointed the key variables for the construction of the vine copula
model, fitted the marginal distribution functions for multiple variables, and constructed the RDV-Copula
functions considering the spatial-temporal relationships. Subsequent to this, a synchronization frequency
analysis based on the copula model was executed to delve deeper into the characteristics of the watershed.
To gauge the efficacy of this method, three benchmark vine copula models, each predicated on different
dimensions and variable relationships, were constructed. Stochastic simulations were then employed to





generate arrays of daily inflow sequences over a typical flood month, with both conditional and
unconditional simulation methods being critically compared. Key findings are summarized below.
(1) The results of our study demonstrated that, within the Shifeng Creek watershed, the synchronization

probability among the four sites reaches up to 41.92%, with the average synchronization probability

between any two sites hitting 65.87%. This strong spatial connectivity indicates a potential for heavy

rainfall events to exacerbate flooding risks downstream.

(2) This study revealed that increasing model dimensions does not inherently improve simulation

outcomes. The high-dimensional copula function, while it can capture more information on the

variables, also makes the structure more complicated. The RDV-Copula method not only ensures

comprehensive data integration but also diminishes the complexity and dimensionality of the vine

copula function, showcasing an optimal balance between information accuracy and model simplicity.

(3) The conditional simulation is a double-edged sword. In comparison to unconditional simulation, for

temporally correlated runoff sequences, conditional simulation can better follow the properties of

prior conditions. However, with an increase in the copula's dimensionality, relying on previously

simulated runoff as a basis for current day predictions can accumulate errors, reducing the overall

simulation accuracy.

In summary, our proposed approach can effectively consolidate relevant spatial-temporal

information for multisite runoff simulations, striking a critical balance between detailed representation
and practical model complexity. This methodology enhances the applicability of vine copula models for
analyzing and managing flood risks. The results obtained using this method can provide valuable decision
support for flood control and scheduling, effectively mitigating flood risk.

**Appendix A**
**Table A1 Common hydrological distribution functions**

| Distribution name | Probability distribution function | Parameters |
|---|---|---|
| Gamma distribution (gamma) | $f(x) = \dfrac{x^{k-1}}{\alpha^k(k)} exp\left[\dfrac{-(x)}{\alpha}\right]$ | $k$ - shape parameter ($k > 0$) <br> $\alpha$ – scale parameter ($\alpha > 0$) |





| Distribution | Formula | Parameters |
|---|---|---|
| Exponential distribution (exp) | $f(x) = \begin{cases} \lambda exp(-\lambda x) & , x \geq 0 \\ 0 & , x < 0 \end{cases}$ | $\lambda$ - rate parameter |
| Pearson-III distribution (p3) | $f(x) = \dfrac{\beta^\alpha}{\Gamma(\alpha)}(x - \gamma)^{\alpha-1} e^{-\beta(x-\gamma)}$ | $\alpha$ − shape parameter ($\alpha > 0$) <br> $\beta$ − scale parameter ($\beta > 0$) <br> $\gamma$ − location parameter |
| Generalized extreme value distribution (gev) | $f(x) = exp\left\{-\left(1 + \xi\dfrac{x - \mu}{\alpha}\right)^{-\frac{1}{\xi}}\right\}$ | $\alpha$ − scale parameter ($\alpha > 0$) <br> $\mu$ − location parameter <br> $\xi$ − shape parameter |
| Inverse gaussian distribution (invgauss) | $f(x) = \sqrt{\dfrac{\lambda}{2\pi x^3}}\, exp\left\{\dfrac{-\lambda(x - \mu)^2}{2\mu^2 x}\right\}$ | $\mu$ − mean (location parameter) <br> $\lambda$ − shape parameter |
| Normal distribution (norm) | $f(x) = \dfrac{1}{\sqrt{2\pi}\sigma} exp\left(-\dfrac{(x - \mu)^2}{2\sigma^2}\right)$ | $\mu$ − location parameter <br> $\sigma$ − scale parameter |
| Logistic distribution (logis) | $f(x) = \dfrac{e^{-(x-\mu)/\gamma}}{\gamma(1 + e^{-(x-\mu)/\gamma})^2}$ | $\mu$ − location parameter <br> $\gamma$ − shape parameter ($\gamma > 0$) |
| Log-normal distribution (lnorm) | $f(x) = \begin{cases} \dfrac{1}{x\sqrt{2\pi}\sigma} exp\left[-\dfrac{1}{2\sigma^2}(lnx - \mu)^2\right] & , x > 0 \\ 0 & , x \leq 0 \end{cases}$ | $\mu$ − location parameter <br> $\sigma$ − scale parameter |
| Log-logistic distribution (llogis) | $f(x) = \dfrac{\left(\frac{\beta}{\alpha}\right)\frac{x^{\beta-1}}{\alpha}}{\left[1 + \left(\frac{x}{\alpha}\right)^\beta\right]^2}\, , x > 0$ | $\alpha$ − scale parameter ($\alpha > 0$) <br> $\beta$ − shape parameter ($\beta > 0$) |
| Generalized pareto distribution (gpd) | $f(x) = \dfrac{1}{\sigma}\left(1 + k\dfrac{(x - \mu)}{\sigma}\right)^{-1-1/k}$ | $\mu$ − location parameter <br> $\sigma$ − scale parameter <br> $k$ - shape parameter |
| Weibull distribution (weibull) | $f(x) = \dfrac{k}{\alpha}\left(\dfrac{x - \gamma}{\alpha}\right)^{k-1} exp\left[-\left(\dfrac{x - \gamma}{\alpha}\right)^k\right]$ | $k$ - shape parameter ($k > 0$) <br> $\alpha$ − scale parameter ($\alpha > 0$) <br> $\gamma$ − location parameter |
| Gumbel distribution (gumbel) | $f(x) = \dfrac{1}{\sigma} exp\left(-\dfrac{x - \mu}{\sigma} - exp\left(-\dfrac{x - \mu}{\sigma}\right)\right)$ | $\mu$ − location parameter <br> $\sigma$ − scale parameter |




**Appendix B**

The probability formulas for the 81 combinations are presented as follows.

(1)   The probability of Type [X-H, Y-H, Z-H, W-H] is as follows:

$$P(X > X_{ph}, Y > Y_{ph}, Z > Z_{ph}, W > W_{ph}) = 1 - u_{ph} - v_{ph} - r_{ph} - s_{ph}$$
$$+ C(u_{ph}, v_{ph}) + C(u_{ph}, r_{ph}) + C(u_{ph}, s_{ph}) + C(v_{ph}, r_{ph}) + C(v_{ph}, s_{ph})$$
$$+ C(r_{ph}, s_{ph}) - C(u_{ph}, v_{ph}, r_{ph}) - C(u_{ph}, v_{ph}, s_{ph}) - C(u_{ph}, r_{ph}, s_{ph})$$
$$- C(v_{ph}, r_{ph}, s_{ph}) + C(u_{ph}, v_{ph}, r_{ph}, s_{ph})$$

(2)   The probability of Type [X-M, Y-M, Z-M, W-M] is as follows:

$$P = (X_{pl} < X < X_{ph}, Y_{pl} < Y < Y_{ph}, Z_{pl} < Z < Z_{ph}, W_{pl} < W < W_{ph})$$
$$= C(u_{ph}, v_{ph}, r_{ph}, s_{ph}) - C(u_{ph}, v_{ph}, r_{ph}, s_{pl}) - C(u_{ph}, v_{ph}, r_{pl}, s_{ph})$$
$$- C(u_{ph}, v_{pl}, r_{ph}, s_{ph}) - C(u_{pl}, v_{ph}, r_{ph}, s_{ph}) + C(u_{ph}, v_{ph}, r_{pl}, s_{pl})$$
$$+ C(u_{ph}, v_{pl}, r_{ph}, s_{pl}) + C(u_{pl}, v_{ph}, r_{ph}, s_{pl}) + C(u_{ph}, v_{pl}, r_{pl}, s_{ph})$$
$$+ C(u_{pl}, v_{ph}, r_{pl}, s_{ph}) + C(u_{pl}, v_{pl}, r_{ph}, s_{ph}) - C(u_{ph}, v_{pl}, r_{pl}, s_{pl})$$
$$- C(u_{pl}, v_{ph}, r_{pl}, s_{pl}) - C(u_{pl}, v_{pl}, r_{ph}, s_{pl}) - C(u_{pl}, v_{pl}, r_{pl}, s_{ph})$$
$$+ C(u_{pl}, v_{pl}, r_{pl}, s_{pl})$$

(3)   The probability of Type [X-L, Y-L, Z-L, W-L] is as follows:

$$P(X < X_{pl}, Y < Y_{pl}, Z < Z_{pl}, W < W_{pl}) = C(u_{pl}, v_{pl}, r_{pl}, s_{pl})$$

(4)   The probability of Type [X-L, Y-H, Z-H, W-H] is as follows:

$$P(X < X_{pl}, Y > Y_{ph}, Z > Z_{ph}, W > W_{ph}) = u_{pl} - C(u_{pl}, v_{ph}) - C(u_{pl}, r_{ph})$$
$$- C(u_{pl}, s_{ph}) + C(u_{pl}, v_{ph}, r_{ph}) + C(u_{pl}, v_{ph}, s_{ph}) + C(u_{pl}, r_{ph}, s_{ph})$$
$$- C(u_{pl}, v_{ph}, r_{ph}, s_{ph})$$

(5)   The probability of Type [X-H, Y-L, Z-H, W-H] is as follows:

$$P(X > X_{ph}, Y < Y_{pl}, Z > Z_{ph}, W > W_{ph}) = v_{pl} - C(u_{ph}, v_{pl}) - C(v_{pl}, r_{ph})$$
$$- C(v_{pl}, s_{ph}) + C(u_{ph}, v_{pl}, r_{ph}) + C(u_{ph}, v_{pl}, s_{ph}) + C(v_{pl}, r_{ph}, s_{ph})$$
$$- C(u_{ph}, v_{pl}, r_{ph}, s_{ph})$$

(6)   The probability of Type [X-H, Y-H, Z-L, W-H] is as follows:

$$P(X > X_{ph}, Y > Y_{ph}, Z < Z_{pl}, W > W_{ph}) = r_{pl} - C(u_{ph}, r_{pl}) - C(v_{ph}, r_{pl})$$
$$- C(r_{pl}, s_{ph}) + C(u_{ph}, v_{ph}, r_{pl}) + C(u_{ph}, r_{pl}, s_{ph}) + C(v_{ph}, r_{pl}, s_{ph})$$
$$- C(u_{ph}, v_{ph}, r_{pl}, s_{ph})$$

(7)   The probability of Type [X-H, Y-H, Z-H, W-L] is as follows:

$$P(X > X_{ph}, Y > Y_{ph}, Z > Z_{ph}, W < W_{pl}) = s_{pl} - C(u_{ph}, s_{pl}) - C(v_{ph}, s_{pl})$$
$$- C(r_{ph}, s_{pl}) + C(u_{ph}, v_{ph}, s_{pl}) + C(u_{ph}, r_{ph}, s_{pl}) + C(v_{ph}, r_{ph}, s_{pl})$$
$$- C(u_{ph}, v_{ph}, r_{ph}, s_{pl})$$

(8)   The probability of Type [X-M, Y-H, Z-H, W-H] is as follows:




$$P(X_{pl} < X < X_{ph}, Y > Y_{ph}, Z > Z_{ph}, W > W_{ph}) = u_{ph} - u_{pl} - C(u_{ph}, v_{ph})$$
$$-C(u_{ph}, r_{ph}) - C(u_{ph}, s_{ph}) + C(u_{pl}, v_{ph}) + C(u_{pl}, r_{ph}) + C(u_{pl}, s_{ph})$$
$$+C(u_{ph}, v_{ph}, r_{ph}) + C(u_{ph}, v_{ph}, s_{ph}) + C(u_{ph}, r_{ph}, s_{ph}) - C(u_{pl}, v_{ph}, r_{ph})$$
$$-C(u_{pl}, v_{ph}, s_{ph}) - C(u_{pl}, r_{ph}, s_{ph}) - C(u_{ph}, v_{ph}, r_{ph}, s_{ph})$$
$$+C(u_{pl}, v_{ph}, r_{ph}, s_{ph})$$


(9) The probability of Type [X-H, Y-M, Z-H, W-H] is as follows:

$$P(X > X_{ph}, Y_{pl} < Y < Y_{ph}, Z > Z_{ph}, W > W_{ph}) = v_{ph} - v_{pl} - C(u_{ph}, v_{ph})$$
$$-C(v_{ph}, r_{ph}) - C(v_{ph}, s_{ph}) + C(u_{ph}, v_{pl}) + C(v_{pl}, r_{ph}) + C(v_{pl}, s_{ph})$$
$$+C(u_{ph}, v_{ph}, r_{ph}) + C(u_{ph}, v_{ph}, s_{ph}) + C(v_{ph}, r_{ph}, s_{ph}) - C(u_{ph}, v_{pl}, r_{ph})$$
$$-C(u_{ph}, v_{pl}, s_{ph}) - C(v_{pl}, r_{ph}, s_{ph}) - C(u_{ph}, v_{ph}, r_{ph}, s_{ph})$$
$$+C(u_{ph}, v_{pl}, r_{ph}, s_{ph})$$


(10) The probability of Type [X-H, Y-H, Z-M, W-H] is as follows:

$$P(X > X_{ph}, Y > Y_{ph}, Z_{pl} < Z < Z_{ph}, W > W_{ph}) = r_{ph} - r_{pl} - C(u_{ph}, r_{ph})$$
$$-C(v_{ph}, r_{ph}) - C(r_{ph}, s_{ph}) + C(u_{ph}, r_{pl}) + C(v_{ph}, r_{pl}) + C(r_{pl}, s_{ph})$$
$$+C(u_{ph}, v_{ph}, r_{ph}) + C(u_{ph}, r_{ph}, s_{ph}) + C(v_{ph}, r_{ph}, s_{ph}) - C(u_{ph}, v_{ph}, r_{pl})$$
$$-C(u_{ph}, r_{pl}, s_{ph}) - C(v_{ph}, r_{pl}, s_{ph}) - C(u_{ph}, v_{ph}, r_{ph}, s_{ph})$$
$$+C(u_{ph}, v_{ph}, r_{pl}, s_{ph})$$


(11) The probability of Type [X-H, Y-H, Z-H, W-M] is as follows:

$$P(X > X_{ph}, Y > Y_{ph}, Z > Z_{ph}, W_{pl} < W < W_{ph}) = s_{ph} - s_{pl} - C(u_{ph}, s_{ph})$$
$$-C(v_{ph}, s_{ph}) - C(r_{ph}, s_{ph}) + C(u_{ph}, s_{pl}) + C(v_{ph}, s_{pl}) + C(r_{ph}, s_{pl})$$
$$+C(u_{ph}, v_{ph}, s_{ph}) + C(u_{ph}, r_{ph}, s_{ph}) + C(v_{ph}, r_{ph}, s_{ph}) - C(u_{ph}, v_{ph}, s_{pl})$$
$$-C(u_{ph}, r_{ph}, s_{pl}) - C(v_{ph}, r_{ph}, s_{pl}) - C(u_{ph}, v_{ph}, r_{ph}, s_{ph})$$
$$+C(u_{ph}, v_{ph}, r_{ph}, s_{pl})$$


(12) The probability of Type [X-L, Y-L, Z-H, W-H] is as follows:

$$P(X < X_{pl}, Y < Y_{pl}, Z > Z_{ph}, W > W_{ph}) = C(u_{pl}, v_{pl}) - C(u_{pl}, v_{pl}, r_{ph})$$
$$-C(u_{pl}, v_{pl}, s_{ph}) + C(u_{pl}, v_{pl}, r_{ph}, s_{ph})$$


(13) The probability of Type [X-L, Y-H, Z-L, W-H] is as follows:

$$P(X < X_{pl}, Y > Y_{ph}, Z < Z_{pl}, W > W_{ph}) = C(u_{pl}, r_{pl}) - C(u_{pl}, v_{ph}, r_{pl})$$
$$-C(u_{pl}, r_{pl}, s_{ph}) + C(u_{pl}, v_{ph}, r_{pl}, s_{ph})$$


(14) The probability of Type [X-L, Y-H, Z-H, W-L] is as follows:

$$P(X < X_{pl}, Y > Y_{ph}, Z > Z_{ph}, W < W_{pl}) = C(u_{pl}, s_{pl}) - C(u_{pl}, v_{ph}, s_{pl})$$
$$-C(u_{pl}, r_{ph}, s_{pl}) + C(u_{pl}, v_{ph}, r_{ph}, s_{pl})$$


(15) The probability of Type [X-H, Y-L, Z-L, W-H] is as follows:

$$P(X > X_{ph}, Y < Y_{pl}, Z < Z_{pl}, W > W_{ph}) = C(v_{pl}, r_{pl}) - C(u_{ph}, v_{pl}, r_{pl})$$
$$-C(v_{pl}, r_{pl}, s_{ph}) + C(u_{ph}, v_{pl}, r_{pl}, s_{ph})$$


(16) The probability of Type [X-H, Y-L, Z-H, W-L] is as follows:

$$P(X > X_{ph}, Y < Y_{pl}, Z > Z_{ph}, W < W_{pl}) = C(v_{pl}, s_{pl}) - C(u_{ph}, v_{pl}, s_{pl})$$
$$-C(v_{pl}, r_{ph}, s_{pl}) + C(u_{ph}, v_{pl}, r_{ph}, s_{pl})$$


(17) The probability of Type [X-H, Y-H, Z-L, W-L] is as follows:

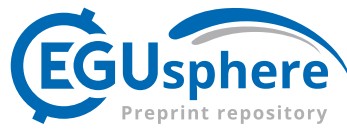

$$P(X > X_{ph}, Y > Y_{ph}, Z < Z_{pl}, W < W_{pl}) = C(r_{pl}, s_{pl}) - C(u_{ph}, r_{pl}, s_{pl})$$
$$-C(v_{ph}, r_{pl}, s_{pl}) + C(u_{ph}, v_{ph}, r_{pl}, s_{pl})$$


(18) The probability of Type [X-M, Y-L, Z-H, W-H] is as follows:

$$P(X_{pl} < X < X_{ph}, Y < Y_{pl}, Z > Z_{ph}, W > W_{ph}) = C(u_{ph}, v_{pl}) - C(u_{pl}, v_{pl})$$
$$-C(u_{ph}, v_{pl}, r_{ph}) - C(u_{ph}, v_{pl}, s_{ph}) + C(u_{pl}, v_{pl}, r_{ph}) + C(u_{pl}, v_{pl}, s_{ph})$$
$$+C(u_{ph}, v_{pl}, r_{ph}, s_{ph}) - C(u_{pl}, v_{pl}, r_{ph}, s_{ph})$$


(19) The probability of Type [X-L, Y-M, Z-H, W-H] is as follows:

$$P(X < X_{pl}, Y_{pl} < Y < Y_{ph}, Z > Z_{ph}, W > W_{ph}) = C(u_{pl}, v_{ph}) - C(u_{pl}, v_{pl})$$
$$-C(u_{pl}, v_{ph}, r_{ph}) - C(u_{pl}, v_{ph}, s_{ph}) + C(u_{pl}, v_{pl}, r_{ph}) + C(u_{pl}, v_{pl}, s_{ph})$$
$$+C(u_{pl}, v_{ph}, r_{ph}, s_{ph}) - C(u_{pl}, v_{pl}, r_{ph}, s_{ph})$$


(20) The probability of Type [X-M, Y-H, Z-L, W-H] is as follows:

$$P(X_{pl} < X < X_{ph}, Y > Y_{ph}, Z < Z_{pl}, W > W_{ph}) = C(u_{ph}, r_{pl}) - C(u_{pl}, r_{pl})$$
$$-C(u_{ph}, v_{ph}, r_{pl}) - C(u_{ph}, r_{pl}, s_{ph}) + C(u_{pl}, v_{ph}, r_{pl}) + C(u_{pl}, r_{pl}, s_{ph})$$
$$+C(u_{ph}, v_{ph}, r_{pl}, s_{ph}) - C(u_{pl}, v_{ph}, r_{pl}, s_{ph})$$


(21) The probability of Type [X-L, Y-H, Z-M, W-H] is as follows:

$$P(X < X_{pl}, Y > Y_{ph}, Z_{pl} < Z < Z_{ph}, W > W_{ph}) = C(u_{pl}, r_{ph}) - C(u_{pl}, r_{pl})$$
$$-C(u_{pl}, v_{ph}, r_{ph}) - C(u_{pl}, r_{ph}, s_{ph}) + C(u_{pl}, v_{ph}, r_{pl}) + C(u_{pl}, r_{pl}, s_{ph})$$
$$+C(u_{pl}, v_{ph}, r_{ph}, s_{ph}) - C(u_{pl}, v_{ph}, r_{pl}, s_{ph})$$


(22) The probability of Type [X-M, Y-H, Z-H, W-L] is as follows:

$$P(X_{pl} < X < X_{ph}, Y > Y_{ph}, Z > Z_{ph}, W < W_{pl}) = C(u_{ph}, s_{pl}) - C(u_{pl}, s_{pl})$$
$$-C(u_{ph}, v_{ph}, s_{pl}) - C(u_{ph}, r_{ph}, s_{pl}) + C(u_{pl}, v_{ph}, s_{pl}) + C(u_{pl}, r_{ph}, s_{pl})$$
$$+C(u_{ph}, v_{ph}, r_{ph}, s_{pl}) - C(u_{pl}, v_{ph}, r_{ph}, s_{pl})$$


(23) The probability of Type [X-L, Y-H, Z-H, W-M] is as follows:

$$P(X < X_{pl}, Y > Y_{ph}, Z > Z_{ph}, W_{pl} < W < W_{ph}) = C(u_{pl}, s_{ph}) - C(u_{pl}, s_{pl})$$
$$-C(u_{pl}, v_{ph}, s_{ph}) - C(u_{pl}, r_{ph}, s_{ph}) + C(u_{pl}, v_{ph}, s_{pl}) + C(u_{pl}, r_{ph}, s_{pl})$$
$$+C(u_{pl}, v_{ph}, r_{ph}, s_{ph}) - C(u_{pl}, v_{ph}, r_{ph}, s_{pl})$$


(24) The probability of Type [X-H, Y-M, Z-L, W-H] is as follows:

$$P(X > X_{ph}, Y_{pl} < Y < Y_{ph}, Z < Z_{pl}, W > W_{ph}) = C(v_{ph}, r_{pl}) - C(v_{pl}, r_{pl})$$
$$-C(u_{ph}, v_{ph}, r_{pl}) - C(v_{ph}, r_{pl}, s_{ph}) + C(u_{ph}, v_{pl}, r_{pl}) + C(v_{pl}, r_{pl}, s_{ph})$$
$$+C(u_{ph}, v_{ph}, r_{pl}, s_{ph}) - C(u_{ph}, v_{pl}, r_{pl}, s_{ph})$$


(25) The probability of Type [X-H, Y-L, Z-M, W-H] is as follows:

$$P(X > X_{ph}, Y < Y_{pl}, Z_{pl} < Z < Z_{ph}, W > W_{ph}) = C(v_{pl}, r_{ph}) - C(v_{pl}, r_{pl})$$
$$-C(u_{ph}, v_{pl}, r_{ph}) - C(v_{pl}, r_{ph}, s_{ph}) + C(u_{ph}, v_{pl}, r_{pl}) + C(v_{pl}, r_{pl}, s_{ph})$$
$$+C(u_{ph}, v_{pl}, r_{ph}, s_{ph}) - C(u_{ph}, v_{pl}, r_{pl}, s_{ph})$$


(26) The probability of Type [X-H, Y-M, Z-H, W-L] is as follows:

$$P(X > X_{ph}, Y_{pl} < Y < Y_{ph}, Z > Z_{ph}, W < W_{pl}) = C(v_{ph}, s_{pl}) - C(v_{pl}, s_{pl})$$
$$-C(u_{ph}, v_{ph}, s_{pl}) - C(v_{ph}, r_{ph}, s_{pl}) + C(u_{ph}, v_{pl}, s_{pl}) + C(v_{pl}, r_{ph}, s_{pl})$$
$$+C(u_{ph}, v_{ph}, r_{ph}, s_{pl}) - C(u_{ph}, v_{pl}, r_{ph}, s_{pl})$$


(27) The probability of Type [X-H, Y-L, Z-H, W-M] is as follows:





$$P(X > X_{ph}, Y < Y_{pl}, Z > Z_{ph}, W_{pl} < W < W_{ph}) = C(v_{pl}, s_{ph}) - C(v_{pl}, s_{pl})$$
$$-C(u_{ph}, v_{pl}, s_{ph}) - C(v_{pl}, r_{ph}, s_{ph}) + C(u_{ph}, v_{pl}, s_{pl}) + C(v_{pl}, r_{ph}, s_{pl})$$
$$+C(u_{ph}, v_{pl}, r_{ph}, s_{ph}) - C(u_{ph}, v_{pl}, r_{ph}, s_{pl})$$

(28) The probability of Type [X-H, Y-H, Z-M, W-L] is as follows:

$$P(X > X_{ph}, Y > Y_{ph}, Z_{pl} < Z < Z_{ph}, W < W_{pl}) = C(r_{ph}, s_{pl}) - C(r_{pl}, s_{pl})$$
$$-C(u_{ph}, r_{ph}, s_{pl}) - C(v_{ph}, r_{ph}, s_{pl}) + C(u_{ph}, r_{pl}, s_{pl}) + C(v_{ph}, r_{pl}, s_{pl})$$
$$+C(u_{ph}, v_{ph}, r_{ph}, s_{pl}) - C(u_{ph}, v_{ph}, r_{pl}, s_{pl})$$

(29) The probability of Type [X-H, Y-H, Z-L, W-M] is as follows:

$$P(X > X_{ph}, Y > Y_{ph}, Z < Z_{pl}, W_{pl} < W < W_{ph}) = C(r_{pl}, s_{ph}) - C(r_{pl}, s_{pl})$$
$$-C(u_{ph}, r_{pl}, s_{ph}) - C(v_{ph}, r_{pl}, s_{ph}) + C(u_{ph}, r_{pl}, s_{pl}) + C(v_{ph}, r_{pl}, s_{pl})$$
$$+C(u_{ph}, v_{ph}, r_{pl}, s_{ph}) - C(u_{ph}, v_{ph}, r_{pl}, s_{pl})$$

(30) The probability of Type [X-M, Y-M, Z-H, W-H] is as follows:

$$P(X_{pl} < X < X_{ph}, Y_{pl} < Y < Y_{ph}, Z > Z_{ph}, W > W_{ph}) = C(u_{ph}, v_{ph})$$
$$+C(u_{pl}, v_{pl}) - C(u_{ph}, v_{pl}) - C(u_{pl}, v_{ph}) - C(u_{ph}, v_{ph}, r_{ph})$$
$$-C(u_{ph}, v_{ph}, s_{ph}) + C(u_{pl}, v_{ph}, r_{ph}) + C(u_{pl}, v_{ph}, s_{ph}) + C(u_{ph}, v_{pl}, r_{ph})$$
$$+C(u_{ph}, v_{pl}, s_{ph}) - C(u_{pl}, v_{pl}, r_{ph}) - C(u_{pl}, v_{pl}, s_{ph}) + C(u_{ph}, v_{ph}, r_{ph}, s_{ph})$$
$$-C(u_{pl}, v_{ph}, r_{ph}, s_{ph}) - C(u_{ph}, v_{pl}, r_{ph}, s_{ph}) + C(u_{pl}, v_{pl}, r_{ph}, s_{ph})$$

(31) The probability of Type [X-M, Y-H, Z-M, W-H] is as follows:

$$P(X_{pl} < X < X_{ph}, Y > Y_{ph}, Z_{pl} < Z < Z_{ph}, W > W_{ph}) = C(u_{ph}, r_{ph})$$
$$+C(u_{pl}, r_{pl}) - C(u_{ph}, r_{pl}) - C(u_{pl}, r_{ph}) - C(u_{ph}, v_{ph}, r_{ph})$$
$$-C(u_{ph}, r_{ph}, s_{ph}) + C(u_{pl}, v_{ph}, r_{ph}) + C(u_{pl}, r_{ph}, s_{ph}) + C(u_{ph}, v_{ph}, r_{pl})$$
$$+C(u_{ph}, r_{pl}, s_{ph}) - C(u_{pl}, v_{ph}, r_{pl}) - C(u_{pl}, r_{pl}, s_{ph}) + C(u_{ph}, v_{ph}, r_{ph}, s_{ph})$$
$$-C(u_{pl}, v_{ph}, r_{ph}, s_{ph}) - C(u_{ph}, v_{ph}, r_{pl}, s_{ph}) + C(u_{pl}, v_{ph}, r_{pl}, s_{ph})$$

(32) The probability of Type [X-M, Y-H, Z-H, W-M] is as follows:

$$P(X_{pl} < X < X_{ph}, Y > Y_{ph}, Z > Z_{ph}, W_{pl} < W < W_{ph}) = C(u_{ph}, s_{ph})$$
$$+C(u_{pl}, s_{pl}) - C(u_{ph}, s_{pl}) - C(u_{pl}, s_{ph}) - C(u_{ph}, v_{ph}, s_{ph})$$
$$-C(u_{ph}, r_{ph}, s_{ph}) + C(u_{pl}, v_{ph}, s_{ph}) + C(u_{pl}, r_{ph}, s_{ph}) + C(u_{ph}, v_{ph}, s_{pl})$$
$$+C(u_{ph}, r_{ph}, s_{pl}) - C(u_{pl}, v_{ph}, s_{pl}) - C(u_{pl}, r_{ph}, s_{pl}) + C(u_{ph}, v_{ph}, r_{ph}, s_{ph})$$
$$-C(u_{pl}, v_{ph}, r_{ph}, s_{ph}) - C(u_{ph}, v_{ph}, r_{ph}, s_{pl}) + C(u_{pl}, v_{ph}, r_{ph}, s_{pl})$$

(33) The probability of Type [X-H, Y-M, Z-M, W-H] is as follows:

$$P(X > X_{ph}, Y_{pl} < Y < Y_{ph}, Z_{pl} < Z < Z_{ph}, W > W_{ph}) = C(v_{ph}, r_{ph})$$
$$+C(v_{pl}, r_{pl}) - C(v_{ph}, r_{pl}) - C(v_{pl}, r_{ph}) - C(u_{ph}, v_{ph}, r_{ph})$$
$$-C(v_{ph}, r_{ph}, s_{ph}) + C(u_{ph}, v_{pl}, r_{ph}) + C(v_{pl}, r_{ph}, s_{ph}) + C(u_{ph}, v_{ph}, r_{pl})$$
$$+C(v_{ph}, r_{pl}, s_{ph}) - C(u_{ph}, v_{pl}, r_{pl}) - C(v_{pl}, r_{pl}, s_{ph}) + C(u_{ph}, v_{ph}, r_{ph}, s_{ph})$$
$$-C(u_{ph}, v_{pl}, r_{ph}, s_{ph}) - C(u_{ph}, v_{ph}, r_{pl}, s_{ph}) + C(u_{ph}, v_{pl}, r_{pl}, s_{ph})$$

(34) The probability of Type [X-H, Y-M, Z-H, W-M] is as follows:



$$P(X > X_{ph}, Y_{pl} < Y < Y_{ph}, Z > Z_{ph}, W_{pl} < W < W_{ph}) = C(v_{ph}, s_{ph})$$
$$+C(v_{pl}, s_{pl}) - C(v_{ph}, s_{pl}) - C(v_{pl}, s_{ph}) - C(u_{ph}, v_{ph}, s_{ph})$$
$$-C(v_{ph}, r_{ph}, s_{ph}) + C(u_{ph}, v_{pl}, s_{ph}) + C(v_{pl}, r_{ph}, s_{ph}) + C(u_{ph}, v_{ph}, s_{pl})$$
$$+C(v_{ph}, r_{ph}, s_{pl}) - C(u_{ph}, v_{pl}, s_{pl}) - C(v_{pl}, r_{ph}, s_{pl}) + C(u_{ph}, v_{ph}, r_{ph}, s_{ph})$$
$$-C(u_{ph}, v_{pl}, r_{ph}, s_{ph}) - C(u_{ph}, v_{ph}, r_{ph}, s_{pl}) + C(u_{ph}, v_{pl}, r_{ph}, s_{pl})$$

(35) The probability of Type [X-H, Y-H, Z-M, W-M] is as follows:

$$P(X > X_{ph}, Y > Y_{ph}, Z_{pl} < Z < Z_{ph}, W_{pl} < W < W_{ph}) = C(r_{ph}, s_{ph})$$
$$+C(r_{pl}, s_{pl}) - C(r_{ph}, s_{pl}) - C(r_{pl}, s_{ph}) - C(u_{ph}, r_{ph}, s_{ph})$$
$$-C(v_{ph}, r_{ph}, s_{ph}) + C(u_{ph}, r_{pl}, s_{ph}) + C(v_{ph}, r_{pl}, s_{ph}) + C(u_{ph}, r_{ph}, s_{pl})$$
$$+C(v_{ph}, r_{ph}, s_{pl}) - C(u_{ph}, r_{pl}, s_{pl}) - C(v_{ph}, r_{pl}, s_{pl}) + C(u_{ph}, v_{ph}, r_{ph}, s_{ph})$$
$$-C(u_{ph}, v_{ph}, r_{pl}, s_{ph}) - C(u_{ph}, v_{ph}, r_{ph}, s_{pl}) + C(u_{ph}, v_{ph}, r_{pl}, s_{pl})$$

(36) The probability of Type [X-M, Y-M, Z-M, W-H] is as follows:

$$P(X_{pl} < X < X_{ph}, Y_{pl} < Y < Y_{ph}, Z_{pl} < Z < Z_{ph}, W > W_{ph})$$
$$= C(u_{ph}, v_{ph}, r_{ph}) - C(u_{ph}, v_{ph}, r_{pl}) - C(u_{ph}, v_{pl}, r_{ph}) - C(u_{pl}, v_{ph}, r_{ph})$$
$$+C(u_{pl}, v_{pl}, r_{ph}) + C(u_{pl}, v_{ph}, r_{pl}) + C(u_{ph}, v_{pl}, r_{pl}) - C(u_{pl}, v_{pl}, r_{pl})$$
$$-C(u_{ph}, v_{ph}, r_{ph}, s_{ph}) + C(u_{pl}, v_{ph}, r_{ph}, s_{ph}) + C(u_{ph}, v_{pl}, r_{ph}, s_{ph})$$
$$+C(u_{ph}, v_{ph}, r_{pl}, s_{ph}) - C(u_{pl}, v_{pl}, r_{ph}, s_{ph}) - C(u_{pl}, v_{ph}, r_{pl}, s_{ph})$$
$$-C(u_{ph}, v_{pl}, r_{pl}, s_{ph}) + C(u_{pl}, v_{pl}, r_{pl}, s_{ph})$$

(37) The probability of Type [X-H, Y-M, Z-M, W-M] is as follows:

$$P(X > X_{ph}, Y_{pl} < Y < Y_{ph}, Z_{pl} < Z < Z_{ph}, W_{pl} < W < W_{ph})$$
$$= C(v_{ph}, r_{ph}, s_{ph}) - C(v_{ph}, r_{ph}, s_{pl}) - C(v_{ph}, r_{pl}, s_{ph}) - C(v_{pl}, r_{ph}, s_{ph})$$
$$+C(v_{pl}, r_{ph}, s_{pl}) + C(v_{pl}, r_{pl}, s_{ph}) + C(v_{ph}, r_{pl}, s_{pl}) - C(v_{pl}, r_{pl}, s_{pl})$$
$$-C(u_{ph}, v_{ph}, r_{ph}, s_{ph}) + C(u_{ph}, v_{pl}, r_{ph}, s_{ph}) + C(u_{ph}, v_{ph}, r_{pl}, s_{ph})$$
$$+C(u_{ph}, v_{ph}, r_{ph}, s_{pl}) - C(u_{ph}, v_{pl}, r_{pl}, s_{ph}) - C(u_{ph}, v_{pl}, r_{ph}, s_{pl})$$
$$-C(u_{ph}, v_{ph}, r_{pl}, s_{pl}) + C(u_{ph}, v_{pl}, r_{pl}, s_{pl})$$

(38) The probability of Type [X-M, Y-H, Z-M, W-M] is as follows:

$$P(X_{pl} < X < X_{ph}, Y_{pl} < Y < Y_{ph}, Z > Z_{ph}, W_{pl} < W < W_{ph})$$
$$= C(u_{ph}, r_{ph}, s_{ph}) - C(u_{ph}, r_{ph}, s_{pl}) - C(u_{ph}, r_{pl}, s_{ph}) - C(u_{pl}, r_{ph}, s_{ph})$$
$$+C(u_{pl}, r_{ph}, s_{pl}) + C(u_{pl}, r_{pl}, s_{ph}) + C(u_{ph}, r_{pl}, s_{pl}) - C(u_{pl}, r_{pl}, s_{pl})$$
$$-C(u_{ph}, v_{ph}, r_{ph}, s_{ph}) + C(u_{pl}, v_{ph}, r_{ph}, s_{ph}) + C(u_{ph}, v_{ph}, r_{pl}, s_{ph})$$
$$+C(u_{ph}, v_{ph}, r_{ph}, s_{pl}) - C(u_{pl}, v_{ph}, r_{pl}, s_{ph}) - C(u_{pl}, v_{ph}, r_{ph}, s_{pl})$$
$$-C(u_{ph}, v_{ph}, r_{pl}, s_{pl}) + C(u_{pl}, v_{ph}, r_{pl}, s_{pl})$$

(39) The probability of Type [X-M, Y-M, Z-H, W-M] is as follows:

$$P(X_{pl} < X < X_{ph}, Y_{pl} < Y < Y_{ph}, Z > Z_{ph}, W_{pl} < W < W_{ph})$$
$$= C(u_{ph}, v_{ph}, s_{ph}) - C(u_{ph}, v_{ph}, s_{pl}) - C(u_{ph}, v_{pl}, s_{ph}) - C(u_{pl}, v_{ph}, s_{ph})$$
$$+C(u_{ph}, v_{pl}, s_{ph}) + C(u_{pl}, v_{ph}, s_{pl}) + C(u_{ph}, v_{pl}, s_{pl}) - C(u_{pl}, v_{pl}, s_{pl})$$
$$-C(u_{ph}, v_{ph}, r_{ph}, s_{ph}) + C(u_{pl}, v_{ph}, r_{ph}, s_{ph}) + C(u_{ph}, v_{pl}, r_{ph}, s_{ph})$$
$$+C(u_{ph}, v_{ph}, r_{ph}, s_{pl}) - C(u_{pl}, v_{pl}, r_{ph}, s_{ph}) - C(u_{pl}, v_{ph}, r_{ph}, s_{pl})$$
$$-C(u_{ph}, v_{pl}, r_{ph}, s_{pl}) + C(u_{pl}, v_{pl}, r_{ph}, s_{pl})$$

(40) The probability of Type [X-M, Y-M, Z-L, W-H] is as follows:





$$P(X_{pl} < X < X_{ph}, Y_{pl} < Y < Y_{ph}, Z < Z_{pl}, W > W_{ph}) = C(u_{ph}, v_{ph}, r_{pl})$$
$$-C(u_{ph}, v_{pl}, r_{pl}) - C(u_{pl}, v_{ph}, r_{pl}) + C(u_{pl}, v_{pl}, r_{pl}) - C(u_{ph}, v_{ph}, r_{pl}, s_{ph})$$
$$+C(u_{pl}, v_{ph}, r_{pl}, s_{ph}) + C(u_{ph}, v_{pl}, r_{pl}, s_{ph}) - C(u_{pl}, v_{pl}, r_{pl}, s_{ph})$$

(41) The probability of Type [X-M, Y-M, Z-H, W-L] is as follows:

$$P(X_{pl} < X < X_{ph}, Y_{pl} < Y < Y_{ph}, Z > Z_{ph}, W < W_{pl}) = C(u_{ph}, v_{ph}, s_{pl})$$
$$-C(u_{ph}, v_{pl}, s_{pl}) - C(u_{pl}, v_{ph}, s_{pl}) + C(u_{pl}, v_{pl}, s_{pl}) - C(u_{ph}, v_{ph}, r_{ph}, s_{pl})$$
$$+C(u_{pl}, v_{ph}, r_{ph}, s_{pl}) + C(u_{ph}, v_{pl}, r_{ph}, s_{pl}) - C(u_{pl}, v_{pl}, r_{ph}, s_{pl})$$

(42) The probability of Type [X-M, Y-L, Z-M, W-H] is as follows:

$$P(X_{pl} < X < X_{ph}, Y < Y_{pl}, Z_{pl} < Z < Z_{ph}, W > W_{ph}) = C(u_{ph}, v_{pl}, r_{ph})$$
$$-C(u_{pl}, v_{pl}, r_{ph}) - C(u_{ph}, v_{pl}, r_{pl}) + C(u_{pl}, v_{pl}, r_{pl}) - C(u_{ph}, v_{pl}, r_{ph}, s_{ph})$$
$$+C(u_{pl}, v_{pl}, r_{ph}, s_{ph}) + C(u_{ph}, v_{pl}, r_{pl}, s_{ph}) - C(u_{pl}, v_{pl}, r_{pl}, s_{ph})$$

(43) The probability of Type [X-M, Y-H, Z-M, W-L] is as follows:

$$P(X_{pl} < X < X_{ph}, Y > Y_{ph}, Z_{pl} < Z < Z_{ph}, W < W_{pl}) = C(u_{ph}, r_{ph}, s_{pl})$$
$$-C(u_{pl}, r_{ph}, s_{pl}) - C(u_{ph}, r_{pl}, s_{pl}) + C(u_{pl}, r_{pl}, s_{pl}) - C(u_{ph}, v_{ph}, r_{ph}, s_{pl})$$
$$+C(u_{pl}, v_{ph}, r_{ph}, s_{pl}) + C(u_{ph}, v_{ph}, r_{pl}, s_{pl}) - C(u_{pl}, v_{ph}, r_{pl}, s_{pl})$$

(44) The probability of Type [X-M, Y-H, Z-L, W-M] is as follows:

$$P(X_{pl} < X < X_{ph}, Y > Y_{ph}, Z < Z_{pl}, W_{pl} < W < W_{ph}) = C(u_{ph}, r_{pl}, s_{ph})$$
$$-C(u_{pl}, r_{pl}, s_{ph}) - C(u_{ph}, r_{pl}, s_{pl}) + C(u_{pl}, r_{pl}, s_{pl}) - C(u_{ph}, v_{ph}, r_{pl}, s_{ph})$$
$$+C(u_{pl}, v_{ph}, r_{pl}, s_{ph}) + C(u_{ph}, v_{ph}, r_{pl}, s_{pl}) - C(u_{pl}, v_{ph}, r_{pl}, s_{pl})$$

(45) The probability of Type [X-M, Y-L, Z-H, W-M] is as follows:

$$P(X_{pl} < X < X_{ph}, Y < Y_{pl}, Z > Z_{ph}, W_{pl} < W < W_{ph}) = C(u_{ph}, v_{pl}, s_{ph})$$
$$-C(u_{pl}, v_{pl}, s_{ph}) - C(u_{ph}, v_{pl}, s_{pl}) + C(u_{pl}, v_{pl}, s_{pl}) - C(u_{ph}, v_{pl}, r_{ph}, s_{ph})$$
$$+C(u_{pl}, v_{pl}, r_{ph}, s_{ph}) + C(u_{ph}, v_{pl}, r_{ph}, s_{pl}) - C(u_{pl}, v_{pl}, r_{ph}, s_{pl})$$

(46) The probability of Type [X-L, Y-M, Z-M, W-H] is as follows:

$$P(X < X_{pl}, Y_{pl} < Y < Y_{ph}, Z_{pl} < Z < Z_{ph}, W > W_{ph}) = C(u_{pl}, v_{ph}, r_{ph})$$
$$-C(u_{pl}, v_{pl}, r_{ph}) - C(u_{pl}, v_{ph}, r_{pl}) + C(u_{pl}, v_{pl}, r_{pl}) - C(u_{pl}, v_{ph}, r_{ph}, s_{ph})$$
$$+C(u_{pl}, v_{pl}, r_{ph}, s_{ph}) + C(u_{pl}, v_{ph}, r_{pl}, s_{ph}) - C(u_{pl}, v_{pl}, r_{pl}, s_{ph})$$

(47) The probability of Type [X-H, Y-M, Z-M, W-L] is as follows:

$$P(X > X_{ph}, Y_{pl} < Y < Y_{ph}, Z_{pl} < Z < Z_{ph}, W < W_{pl}) = C(v_{ph}, r_{ph}, s_{pl})$$
$$-C(v_{pl}, r_{ph}, s_{pl}) - C(v_{ph}, r_{pl}, s_{pl}) + C(v_{pl}, r_{pl}, s_{pl}) - C(u_{ph}, v_{ph}, r_{ph}, s_{pl})$$
$$+C(u_{ph}, v_{pl}, r_{ph}, s_{pl}) + C(u_{ph}, v_{ph}, r_{pl}, s_{pl}) - C(u_{ph}, v_{pl}, r_{pl}, s_{pl})$$

(48) The probability of Type [X-H, Y-M, Z-L, W-M]] is as follows:

$$P(X > X_{ph}, Y_{pl} < Y < Y_{ph}, Z < Z_{pl}, W_{pl} < W < W_{ph}) = C(v_{ph}, r_{pl}, s_{ph})$$
$$-C(v_{pl}, r_{pl}, s_{ph}) - C(v_{ph}, r_{pl}, s_{pl}) + C(v_{pl}, r_{pl}, s_{pl}) - C(u_{ph}, v_{ph}, r_{pl}, s_{ph})$$
$$+C(u_{ph}, v_{pl}, r_{pl}, s_{ph}) + C(u_{ph}, v_{ph}, r_{pl}, s_{pl}) - C(u_{ph}, v_{pl}, r_{pl}, s_{pl})$$

(49) The probability of Type [X-L, Y-M, Z-H, W-M]] is as follows:

$$P(X < X_{pl}, Y_{pl} < Y < Y_{ph}, Z > Z_{ph}, W_{pl} < W < W_{ph}) = C(u_{pl}, v_{ph}, s_{ph})$$
$$-C(u_{pl}, v_{pl}, s_{ph}) - C(u_{pl}, v_{ph}, s_{pl}) + C(u_{pl}, v_{pl}, s_{pl}) - C(u_{pl}, v_{ph}, r_{ph}, s_{ph})$$
$$+C(u_{pl}, v_{pl}, r_{ph}, s_{ph}) + C(u_{pl}, v_{ph}, r_{ph}, s_{pl}) - C(u_{pl}, v_{pl}, r_{ph}, s_{pl})$$



(50) The probability of Type [X-L, Y-H, Z-M, W-M]] is as follows:

$$P(X < X_{pl}, Y > Y_{ph}, Z_{pl} < Z < Z_{ph}, W_{pl} < W < W_{ph}) = C(u_{pl}, r_{ph}, s_{ph})$$
$$-C(u_{pl}, r_{pl}, s_{ph}) - C(u_{pl}, r_{ph}, s_{pl}) + C(u_{pl}, r_{pl}, s_{pl}) - C(u_{pl}, v_{ph}, r_{ph}, s_{ph})$$
$$+C(u_{pl}, v_{ph}, r_{pl}, s_{ph}) + C(u_{pl}, v_{ph}, r_{ph}, s_{pl}) - C(u_{pl}, v_{ph}, r_{pl}, s_{pl})$$

(51) The probability of Type [X-H, Y-L, Z-M, W-M]] is as follows:

$$P(X > X_{ph}, Y < Y_{pl}, Z_{pl} < Z < Z_{ph}, W_{pl} < W < W_{ph}) = C(v_{pl}, r_{ph}, s_{ph})$$
$$-C(v_{pl}, r_{pl}, s_{ph}) - C(v_{pl}, r_{ph}, s_{pl}) + C(v_{pl}, r_{pl}, s_{pl}) - C(u_{ph}, v_{pl}, r_{ph}, s_{ph})$$
$$+C(u_{ph}, v_{pl}, r_{pl}, s_{ph}) + C(u_{ph}, v_{pl}, r_{ph}, s_{pl}) - C(u_{ph}, v_{pl}, r_{pl}, s_{pl})$$

(52) The probability of Type [X-M, Y-L, Z-L, W-H] is as follows:

$$P(X_{pl} < X < X_{ph}, Y < Y_{pl}, Z < Z_{pl}, W > W_{ph}) = C(u_{ph}, v_{pl}, r_{pl})$$
$$-C(u_{pl}, v_{pl}, r_{pl}) - C(u_{ph}, v_{pl}, r_{pl}, s_{ph}) + C(u_{pl}, v_{pl}, r_{pl}, s_{ph})$$

(53) The probability of Type [X-L, Y-M, Z-L, W-H] is as follows:

$$P(X < X_{pl}, Y_{pl} < Y < Y_{ph}, Z < Z_{pl}, W > W_{ph}) = C(u_{pl}, v_{ph}, r_{pl})$$
$$-C(u_{pl}, v_{pl}, r_{pl}) - C(u_{pl}, v_{ph}, r_{pl}, s_{ph}) + C(u_{pl}, v_{pl}, r_{pl}, s_{ph})$$

(54) The probability of Type [X-L, Y-L, Z-M, W-H] is as follows:

$$P(X < X_{pl}, Y_{pl} < Y < Y_{ph}, Z < Z_{pl}, W > W_{ph}) = C(u_{pl}, v_{pl}, r_{ph})$$
$$-C(u_{pl}, v_{pl}, r_{pl}) - C(u_{pl}, v_{pl}, r_{ph}, s_{ph}) + C(u_{pl}, v_{pl}, r_{pl}, s_{ph})$$

(55) The probability of Type [X-M, Y-L, Z-H, W-L] is as follows:

$$P(X_{pl} < X < X_{ph}, Y < Y_{pl}, Z > Z_{ph}, W < W_{pl}) = C(u_{ph}, v_{pl}, s_{pl})$$
$$-C(u_{pl}, v_{pl}, s_{pl}) - C(u_{ph}, v_{pl}, r_{ph}, s_{pl}) + C(u_{pl}, v_{pl}, r_{ph}, s_{pl})$$

(56) The probability of Type [X-L, Y-M, Z-H, W-L] is as follows:

$$P(X < X_{pl}, Y_{pl} < Y < Y_{ph}, Z > Z_{ph}, W < W_{pl}) = C(u_{pl}, v_{ph}, s_{pl})$$
$$-C(u_{pl}, v_{pl}, s_{pl}) - C(u_{pl}, v_{ph}, r_{ph}, s_{pl}) + C(u_{pl}, v_{pl}, r_{ph}, s_{pl})$$

(57) The probability of Type [X-L, Y-L, Z-H, W-M] is as follows:

$$P(X < X_{pl}, Y < Y_{pl}, Z > Z_{ph}, W_{pl} < W < W_{ph}) = C(u_{pl}, v_{pl}, s_{ph})$$
$$-C(u_{pl}, v_{pl}, s_{pl}) - C(u_{pl}, v_{pl}, r_{ph}, s_{ph}) + C(u_{pl}, v_{pl}, r_{ph}, s_{pl})$$

(58) The probability of Type [X-M, Y-H, Z-L, W-L] is as follows:

$$P(X_{pl} < X < X_{ph}, Y > Y_{ph}, Z < Z_{pl}, W < W_{pl}) = C(u_{ph}, r_{pl}, s_{pl})$$
$$-C(u_{pl}, r_{pl}, s_{pl}) - C(u_{ph}, v_{ph}, r_{pl}, s_{pl}) + C(u_{pl}, v_{ph}, r_{pl}, s_{pl})$$

(59) The probability of Type [X-L, Y-H, Z-M, W-L] is as follows:

$$P(X < X_{pl}, Y > Y_{ph}, Z_{pl} < Z < Z_{ph}, W < W_{pl}) = C(u_{pl}, r_{ph}, s_{pl})$$
$$-C(u_{pl}, r_{pl}, s_{pl}) - C(u_{pl}, v_{ph}, r_{ph}, s_{pl}) + C(u_{pl}, v_{ph}, r_{pl}, s_{pl})$$

(60) The probability of Type [X-L, Y-H, Z-L, W-M] is as follows:

$$P(X < X_{pl}, Y > Y_{ph}, Z < Z_{pl}, W_{pl} < W < W_{ph}) = C(u_{pl}, r_{pl}, s_{ph})$$
$$-C(u_{pl}, r_{pl}, s_{pl}) - C(u_{pl}, v_{ph}, r_{pl}, s_{ph}) + C(u_{pl}, v_{ph}, r_{pl}, s_{pl})$$

(61) The probability of Type [X-H, Y-M, Z-L, W-L] is as follows:

$$P(X > X_{ph}, Y_{pl} < Y < Y_{ph}, Z < Z_{pl}, W < W_{pl}) = C(v_{ph}, r_{pl}, s_{pl})$$
$$-C(v_{pl}, r_{pl}, s_{pl}) - C(u_{ph}, v_{ph}, r_{pl}, s_{pl}) + C(u_{ph}, v_{pl}, r_{pl}, s_{pl})$$



(62) The probability of Type [X-H, Y-L, Z-M, W-L] is as follows:
$$P(X > X_{ph}, Y < Y_{pl}, Z_{pl} < Z < Z_{ph}, W < W_{pl}) = C(v_{pl}, r_{ph}, s_{pl})$$
$$-C(v_{pl}, r_{pl}, s_{pl}) - C(u_{ph}, v_{pl}, r_{ph}, s_{pl}) + C(u_{ph}, v_{pl}, r_{pl}, s_{pl})$$

(63) The probability of Type [X-H, Y-L, Z-L, W-M] is as follows:
$$P(X > X_{ph}, Y < Y_{pl}, Z < Z_{pl}, W_{pl} < W < W_{ph}) = C(v_{pl}, r_{pl}, s_{ph})$$
$$-C(v_{pl}, r_{pl}, s_{pl}) - C(u_{ph}, v_{pl}, r_{pl}, s_{ph}) + C(u_{ph}, v_{pl}, r_{pl}, s_{pl})$$

(64) The probability of Type [X-L, Y-L, Z-L, W-H] is as follows:
$$P(X < X_{pl}, Y < Y_{pl}, Z < Z_{pl}, W > W_{ph}) = C(u_{pl}, v_{pl}, r_{pl})$$
$$-C(u_{pl}, v_{pl}, r_{pl}, s_{ph})$$

(65) The probability of Type [X-L, Y-L, Z-H, W-L] is as follows:
$$P(X < X_{pl}, Y < Y_{pl}, Z > Z_{ph}, W < W_{pl}) = C(u_{pl}, v_{pl}, s_{pl})$$
$$-C(u_{pl}, v_{pl}, r_{ph}, s_{pl})$$

(66) The probability of Type [X-L, Y-H, Z-L, W-L] is as follows:
$$P(X < X_{pl}, Y > Y_{ph}, Z < Z_{pl}, W < W_{pl}) = C(u_{pl}, r_{pl}, s_{pl})$$
$$-C(u_{pl}, v_{ph}, r_{pl}, s_{pl})$$

(67) The probability of Type [X-H, Y-L, Z-L, W-L] is as follows:
$$P(X > X_{ph}, Y < Y_{pl}, Z < Z_{pl}, W < W_{pl}) = C(v_{pl}, r_{pl}, s_{pl})$$
$$-C(u_{ph}, v_{pl}, r_{pl}, s_{pl})$$

(68) The probability of Type [X-M, Y-M, Z-M, W-L] is as follows:
$$P(X_{pl} < X < X_{ph}, Y_{pl} < Y < Y_{ph}, Z_{pl} < Z < Z_{ph}, W < W_{pl})$$
$$= C(u_{ph}, v_{ph}, r_{ph}, s_{pl}) - C(u_{ph}, v_{ph}, r_{pl}, s_{pl}) - C(u_{ph}, v_{pl}, r_{ph}, s_{pl})$$
$$-C(u_{pl}, v_{ph}, r_{ph}, s_{pl}) + C(u_{ph}, v_{pl}, r_{pl}, s_{pl}) + C(u_{pl}, v_{ph}, r_{pl}, s_{pl})$$
$$+C(u_{pl}, v_{pl}, r_{ph}, s_{pl}) - C(u_{pl}, v_{pl}, r_{pl}, s_{pl})$$

(69) The probability of Type [X-M, Y-M, Z-L, W-M] is as follows:
$$P(X_{pl} < X < X_{ph}, Y_{pl} < Y < Y_{ph}, Z < Z_{pl}, W_{pl} < W < W_{ph})$$
$$= C(u_{ph}, v_{ph}, r_{pl}, s_{ph}) - C(u_{ph}, v_{ph}, r_{pl}, s_{pl}) - C(u_{ph}, v_{pl}, r_{pl}, s_{ph})$$
$$-C(u_{pl}, v_{ph}, r_{pl}, s_{ph}) + C(u_{ph}, v_{pl}, r_{pl}, s_{pl}) + C(u_{pl}, v_{ph}, r_{pl}, s_{pl})$$
$$+C(u_{pl}, v_{pl}, r_{pl}, s_{ph}) - C(u_{pl}, v_{pl}, r_{pl}, s_{pl})$$

(70) The probability of Type [X-M, Y-L, Z-M, W-M] is as follows:
$$P(X_{pl} < X < X_{ph}, Y < Y_{pl}, Z_{pl} < Z < Z_{ph}, W_{pl} < W < W_{ph})$$
$$= C(u_{ph}, v_{pl}, r_{ph}, s_{ph}) - C(u_{pl}, v_{pl}, r_{ph}, s_{ph}) - C(u_{ph}, v_{pl}, r_{pl}, s_{ph})$$
$$-C(u_{ph}, v_{pl}, r_{ph}, s_{pl}) + C(u_{ph}, v_{pl}, r_{pl}, s_{pl}) + C(u_{pl}, v_{pl}, r_{ph}, s_{pl})$$
$$+C(u_{pl}, v_{pl}, r_{pl}, s_{ph}) - C(u_{pl}, v_{pl}, r_{pl}, s_{pl})$$

(71) The probability of Type [X-L, Y-M, Z-M, W-M] is as follows:
$$P(X < X_{pl}, Y_{pl} < Y < Y_{ph}, Z_{pl} < Z < Z_{ph}, W_{pl} < W < W_{ph})$$
$$= C(u_{pl}, v_{ph}, r_{ph}, s_{ph}) - C(u_{pl}, v_{pl}, r_{ph}, s_{ph}) - C(u_{pl}, v_{ph}, r_{pl}, s_{ph})$$
$$-C(u_{pl}, v_{ph}, r_{ph}, s_{pl}) + C(u_{pl}, v_{ph}, r_{pl}, s_{pl}) + C(u_{pl}, v_{pl}, r_{ph}, s_{pl})$$
$$+C(u_{pl}, v_{pl}, r_{pl}, s_{ph}) - C(u_{pl}, v_{pl}, r_{pl}, s_{pl})$$

(72) The probability of Type [X-M, Y-M, Z-L, W-L] is as follows:



$$P(X_{pl} < X < X_{ph}, Y_{pl} < Y < Y_{ph}, Z < Z_{pl}, W < W_{pl})$$
$$= C(u_{ph}, v_{ph}, r_{pl}, s_{pl}) - C(u_{ph}, v_{pl}, r_{pl}, s_{pl}) - C(u_{pl}, v_{ph}, r_{pl}, s_{pl})$$
$$+C(u_{pl}, v_{pl}, r_{pl}, s_{pl})$$

(73) The probability of Type [X-M, Y-L, Z-M, W-L] is as follows:
$$P(X_{pl} < X < X_{ph}, Y < Y_{pl}, Z_{pl} < Z < Z_{ph}, W < W_{pl})$$
$$= C(u_{ph}, v_{pl}, r_{ph}, s_{pl}) - C(u_{ph}, v_{pl}, r_{pl}, s_{pl}) - C(u_{pl}, v_{pl}, r_{ph}, s_{pl})$$
$$+C(u_{pl}, v_{pl}, r_{pl}, s_{pl})$$

(74) The probability of Type [X-M, Y-L, Z-L, W-M] is as follows:
$$P(X_{pl} < X < X_{ph}, Y < Y_{pl}, Z < Z_{pl}, W_{pl} < W < W_{ph})$$
$$= C(u_{ph}, v_{pl}, r_{pl}, s_{ph}) - C(u_{ph}, v_{pl}, r_{pl}, s_{pl}) - C(u_{pl}, v_{pl}, r_{pl}, s_{ph})$$
$$+C(u_{pl}, v_{pl}, r_{pl}, s_{pl})$$

(75) The probability of Type [X-L, Y-M, Z-M, W-L] is as follows:
$$P(X < X_{pl}, Y_{pl} < Y < Y_{ph}, Z_{pl} < Z < Z_{ph}, W < W_{pl})$$
$$= C(u_{pl}, v_{ph}, r_{ph}, s_{pl}) - C(u_{pl}, v_{ph}, r_{pl}, s_{pl}) - C(u_{pl}, v_{pl}, r_{ph}, s_{pl})$$
$$+C(u_{pl}, v_{pl}, r_{pl}, s_{pl})$$

(76) The probability of Type [X-L, Y-M, Z-L, W-M] is as follows:
$$P(X < X_{pl}, Y_{pl} < Y < Y_{ph}, Z < Z_{pl}, W_{pl} < W < W_{ph})$$
$$= C(u_{pl}, v_{ph}, r_{pl}, s_{ph}) - C(u_{pl}, v_{ph}, r_{pl}, s_{pl}) - C(u_{pl}, v_{pl}, r_{pl}, s_{ph})$$
$$+C(u_{pl}, v_{pl}, r_{pl}, s_{pl})$$

(77) The probability of Type [X-L, Y-L, Z-M, W-M] is as follows:
$$P(X < X_{pl}, Y < Y_{pl}, Z_{pl} < Z < Z_{ph}, W_{pl} < W < W_{ph})$$
$$= C(u_{pl}, v_{pl}, r_{ph}, s_{ph}) - C(u_{pl}, v_{pl}, r_{ph}, s_{pl}) - C(u_{pl}, v_{pl}, r_{pl}, s_{ph})$$
$$+C(u_{pl}, v_{pl}, r_{pl}, s_{pl})$$

(78) The probability of Type [X-M, Y-L, Z-L, W-L] is as follows:
$$P(X_{pl} < X < X_{ph}, Y < Y_{pl}, Z < Z_{pl}, W < W_{pl})$$
$$= C(u_{ph}, v_{pl}, r_{pl}, s_{pl}) - C(u_{pl}, v_{pl}, r_{pl}, s_{pl})$$

(79) The probability of Type [X-L, Y-M, Z-L, W-L] is as follows:
$$P(X < X_{pl}, Y_{pl} < Y < Y_{ph}, Z < Z_{pl}, W < W_{pl})$$
$$= C(u_{pl}, v_{ph}, r_{pl}, s_{pl}) - C(u_{pl}, v_{pl}, r_{pl}, s_{pl})$$

(80) The probability of Type [X-L, Y-L, Z-M, W-L] is as follows:
$$P(X < X_{pl}, Y < Y_{pl}, Z_{pl} < Z < Z_{ph}, W < W_{pl})$$
$$= C(u_{pl}, v_{pl}, r_{ph}, s_{pl}) - C(u_{pl}, v_{pl}, r_{pl}, s_{pl})$$

(81) The probability of Type [X-L, Y-L, Z-L, W-M] is as follows:
$$P(X < X_{pl}, Y < Y_{pl}, Z < Z_{pl}, W_{pl} < W < W_{ph})$$
$$= C(u_{pl}, v_{pl}, r_{pl}, s_{ph}) - C(u_{pl}, v_{pl}, r_{pl}, s_{pl})$$




**Appendix C**

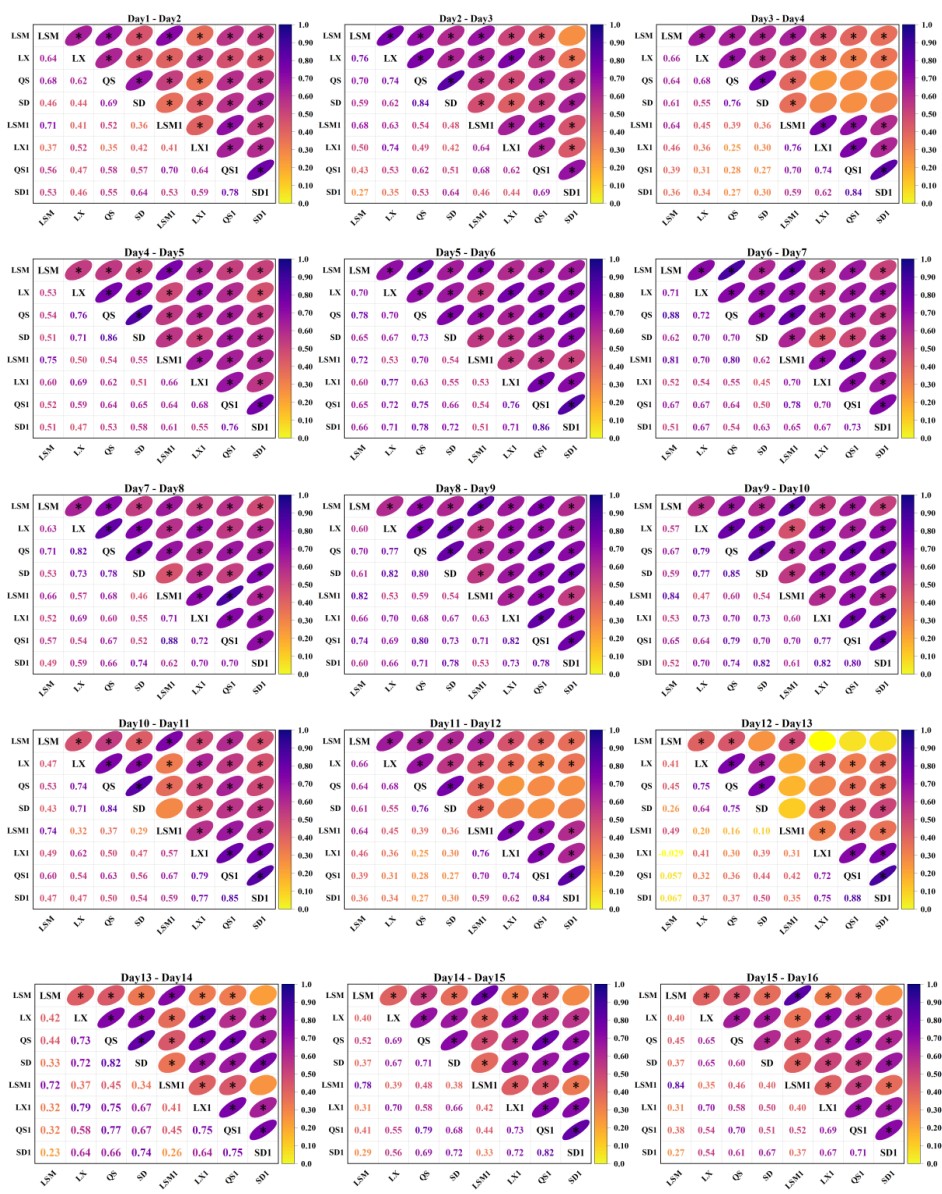





**Figure C1. Results of correlation analysis for daily runoff at multiple sites**



**Appendix D**

A total of twelve different distribution functions were employed to fit the daily runoff flows at the four points for each day in August. For each of the 31 days in August, the preferred marginal distribution functions and their corresponding parameters for each variable can be seen in Table D1. Figure D1 shows the preferred marginal distribution functions for each variable over month of August.

**Table D1 Marginal distributions and parameters preferred for each variable on August 1st-31st**

| Date | variable | distribution | shape | loc | scale | mean | rate | meanlog | sdlog | alpha |
|---|---|---|---|---|---|---|---|---|---|---|
| 1 | LSM | gamma | 0.379 | | | | 0.106 | | | |
| | LX | gev | 0.583 | 0.246 | 0.274 | | | | | |
| | QS | gev | 0.578 | 1.890 | 2.056 | | | | | |
| | SD | gev | 0.643 | 3.716 | 3.670 | | | | | |
| 2 | LSM | gev | 0.539 | 0.854 | 1.434 | | | | | |
| | LX | invgauss | 0.260 | | | 0.715 | | | | |
| | QS | gev | 0.539 | 1.964 | 1.986 | | | | | |
| | SD | llogis | 1.527 | | 5.206 | | | | | |
| 3 | LSM | lnorm | | | | | | -0.437 | 2.817 | |
| | LX | invgauss | 0.182 | | | 1.835 | | | | |
| | QS | lnorm | | | | | | 1.166 | 1.425 | |
| | SD | invgauss | 3.541 | | | 15.295 | | | | |
| 4 | LSM | gev | 0.646 | 1.265 | 2.495 | | | | | |
| | LX | lnorm | | | | | | -0.664 | 1.445 | |
| | QS | gpd | -0.202 | -0.715 | 9.321 | | | | | |
| | SD | gpd | 0.000 | -0.350 | 15.000 | | | | | |
| 5 | LSM | weibull | 0.433 | | 3.195 | | | | | |
| | LX | gev | 0.888 | 0.250 | 0.385 | | | | | |
| | QS | invgauss | 2.133 | | | 8.328 | | | | |
| | SD | gev | 0.626 | 4.946 | 5.406 | | | | | |
| 6 | LSM | gamma | 0.402 | | | | 0.090 | | | |





| | | | | | | | | |
|---|---|---|---|---|---|---|---|---|
| | LX | llogis | 1.277 | | 0.324 | | | |
| | QS | gev | 0.688 | 1.545 | 1.486 | | | |
| | SD | llogis | 1.495 | | 5.761 | | | |
| | LSM | gev | 0.365 | 1.537 | 2.783 | | | |
| 7 | LX | llogis | 1.073 | | 0.459 | | | |
| | QS | lnorm | | | | | 1.072 | 1.567 |
| | SD | gev | 0.836 | 4.670 | 5.745 | | | |
| | LSM | weibull | 0.456 | | 4.064 | | | |
| 8 | LX | invgauss | 0.214 | | | 1.749 | | |
| | QS | llogis | 0.977 | | 3.253 | | | |
| | SD | gpd | 0.846 | -0.712 | 10.057 | | | |
| | LSM | weibull | 0.438 | | 5.072 | | | |
| 9 | LX | invgauss | 0.211 | | | 3.978 | | |
| | QS | lnorm | | | | | 1.368 | 1.887 |
| | SD | lnorm | | | | | 2.433 | 1.905 |
| | LSM | weibull | 0.358 | | 6.476 | | | |
| 10 | LX | lnorm | | | | | -0.005 | 2.051 |
| | QS | lnorm | | | | | 1.678 | 2.274 |
| | SD | lnorm | | | | | 2.720 | 2.410 |
| | LSM | weibull | 0.474 | | 6.926 | | | |
| 11 | LX | lnorm | | | | | 0.127 | 1.718 |
| | QS | lnorm | | | | | 1.899 | 1.923 |
| | SD | llogis | 0.929 | | 16.980 | | | |
| | LSM | llogis | 0.885 | | 1.786 | | | |
| 12 | LX | invgauss | 0.542 | | | 1.797 | | |
| | QS | invgauss | 2.772 | | | 14.129 | | |
| | SD | invgauss | 7.912 | | | 37.729 | | |
| 13 | LSM | gpd | 0.216 | -0.976 | 7.565 | | | |
| | LX | weibull | 0.796 | | 1.774 | | | |



|    |     |          |        |        |        |        |        |       |       |
|----|-----|----------|--------|--------|--------|--------|--------|-------|-------|
|    | QS  | gpd      | 0.299  | -0.095 | 10.472 |        |        |       |       |
|    | SD  | invgauss | 10.011 |        |        | 33.990 |        |       |       |
|    | LSM | gev      | 0.608  | 1.580  | 2.722  |        |        |       |       |
| 14 | LX  | invgauss | 0.432  |        |        | 1.527  |        |       |       |
|    | QS  | invgauss | 3.695  |        |        | 14.640 |        |       |       |
|    | SD  | invgauss | 8.444  |        |        | 31.374 |        |       |       |
|    | LSM | gev      | 0.436  | 1.242  | 2.118  |        |        |       |       |
| 15 | LX  | gumbel   |        |        | 0.655  |        |        |       | 0.515 |
|    | QS  | invgauss | 3.225  |        |        | 7.595  |        |       |       |
|    | SD  | invgauss | 7.520  |        |        | 18.606 |        |       |       |
|    | LSM | weibull  | 0.506  |        | 2.783  |        |        |       |       |
| 16 | LX  | invgauss | 0.360  |        |        | 1.148  |        |       |       |
|    | QS  | invgauss | 2.943  |        |        | 9.336  |        |       |       |
|    | SD  | gpd      | 0.359  | 0.529  | 13.680 |        |        |       |       |
|    | LSM | weibull  | 0.479  |        | 2.907  |        |        |       |       |
| 17 | LX  | weibull  | 0.897  |        | 0.952  |        |        |       |       |
|    | QS  | gpd      | 0.385  | -0.580 | 6.729  |        |        |       |       |
|    | SD  | invgauss | 6.433  |        |        | 19.990 |        |       |       |
|    | LSM | gev      | 0.552  | 1.252  | 2.482  |        |        |       |       |
| 18 | LX  | gev      | 0.492  | 0.411  | 0.493  |        |        |       |       |
|    | QS  | gpd      | 0.300  | -0.632 | 7.393  |        |        |       |       |
|    | SD  | lnorm    |        |        |        |        | 2.290  | 1.315 |       |
|    | LSM | weibull  | 0.452  |        | 3.243  |        |        |       |       |
| 19 | LX  | invgauss | 0.301  |        |        | 1.595  |        |       |       |
|    | QS  | invgauss | 2.268  |        |        | 14.869 |        |       |       |
|    | SD  | gpd      | 0.618  | -0.297 | 11.762 |        |        |       |       |
|    | LSM | lnorm    |        |        |        |        | -0.048 | 2.580 |       |
| 20 | LX  | llogis   | 1.246  |        | 0.593  |        |        |       |       |
|    | QS  | invgauss | 1.989  |        |        | 25.636 |        |       |       |





|    |     |         |        |        |        |        |        |       |
|----|-----|---------|--------|--------|--------|--------|--------|-------|
|    | SD  | gev     | 0.818  | 6.508  | 9.642  |        |        |       |
|    | LSM | gev     | 0.779  | 0.859  | 1.315  |        |        |       |
| 21 | LX  | llogis  | 1.522  |        | 0.528  |        |        |       |
|    | QS  | gev     | 0.738  | 2.163  | 2.485  |        |        |       |
|    | SD  | invgauss| 7.401  |        |        | 27.102 |        |       |
|    | LSM | weibull | 0.521  |        | 2.298  |        |        |       |
| 22 | LX  | llogis  | 1.595  |        | 0.402  |        |        |       |
|    | QS  | invgauss| 2.757  |        |        | 7.322  |        |       |
|    | SD  | invgauss| 7.626  |        |        | 19.094 |        |       |
|    | LSM | weibull | 0.460  |        | 3.114  |        |        |       |
| 23 | LX  | gev     | 0.764  | 0.294  | 0.402  |        |        |       |
|    | QS  | invgauss| 3.491  |        |        | 9.169  |        |       |
|    | SD  | gpd     | 0.345  | 0.923  | 13.719 |        |        |       |
|    | LSM | gev     | 0.619  | 1.204  | 2.195  |        |        |       |
| 24 | LX  | invgauss| 0.293  |        |        | 1.625  |        |       |
|    | QS  | invgauss| 2.790  |        |        | 10.814 |        |       |
|    | SD  | invgauss| 7.810  |        |        | 23.039 |        |       |
|    | LSM | gamma   | 0.438  |        |        |        | 0.073  |       |
| 25 | LX  | gev     | 0.238  | 0.632  | 0.797  |        |        |       |
|    | QS  | gev     | 0.403  | 3.483  | 4.696  |        |        |       |
|    | SD  | gpd     | 0.387  | 0.057  | 14.586 |        |        |       |
|    | LSM | gev     | 0.348  | 2.009  | 3.077  |        |        |       |
| 26 | LX  | weibull | 0.789  |        | 1.674  |        |        |       |
|    | QS  | weibull | 0.759  |        | 11.716 |        |        |       |
|    | SD  | gev     | 0.439  | 12.256 | 17.061 |        |        |       |
|    | LSM | gamma   | 0.533  |        |        |        | 0.127  |       |
| 27 | LX  | lnorm   |        |        |        |        | -0.472 | 1.424 |
|    | QS  | lnorm   |        |        |        |        | 1.549  | 1.321 |
|    | SD  | gev     | 0.555  | 7.945  | 9.853  |        |        |       |



| | | | | | |
|---|---|---|---|---|---|
| 28 | LSM | gev | 0.604 | 1.375 | 2.510 |
| | LX | gev | 0.318 | 0.640 | 0.823 |
| | QS | gev | 0.605 | 3.316 | 4.562 |
| | SD | gpd | 0.328 | -0.247 | 14.191 |
| 29 | LSM | weibull | 0.661 | | 4.721 |
| | LX | gev | -0.186 | 0.938 | 0.851 |
| | QS | gpd | 0.316 | -0.775 | 8.682 |
| | SD | gpd | 0.107 | -0.389 | 17.428 |
| 30 | LSM | gev | 0.699 | 1.338 | 1.895 |
| | LX | gev | 0.547 | 0.500 | 0.639 |
| | QS | invgauss | 3.152 | | 15.179 |
| | SD | gpd | 0.651 | -0.480 | 10.676 |
| 31 | LSM | llogis | 0.868 | | 1.232 |
| | LX | gev | 0.792 | 0.313 | 0.325 |
| | QS | gev | 0.858 | 1.962 | 2.066 |
| | SD | gev | 0.814 | 4.883 | 6.333 |

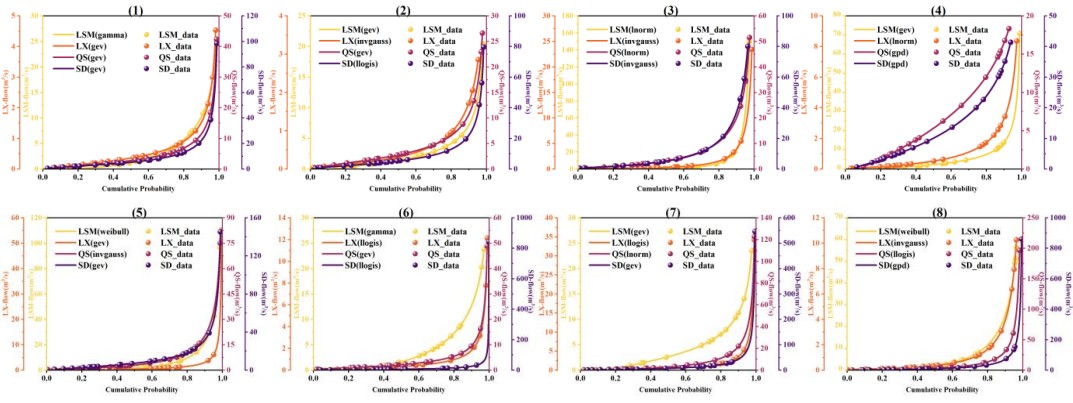



**Figure D1.** **Cumulative probability distribution of the preferred marginal distribution function for runoff on each day throughout August**

**Code availability**

The developed routines for working with conditional joint probability density functions are publicly



available via the rvinecopulib R package (https://github.com/vinecopulib/rvinecopulib) and
CDVineCopulaConditional R package (https://github.com/cran/CDVineCopulaConditional). Other
codes used to support the findings of this study are available from the authors upon request.
**Data Availability**
Streamflow    can    be    checked    form    hydrology    information    of    Taizhou    City    at
http://www.shui00.com/ZhswFloodWater/web/html/index.html?module=wssyq. Other    data    used    to
support the findings of this study are available from the corresponding author upon request.
**Author contribution**
XY and YPX designed the research. HG and SC collected and preprocessed the data. XY and YG
conducted all the experiments and analyzed the results. SC assisted with the paper's background. XY
wrote the first draft of the manuscript with contributions from YPX. YPX supervised the study and edited
the manuscript.
**Competing interests**
At least one of the (co-)authors is a member of the editorial board of Hydrology and Earth System Science.
**Disclaimer**
Copernicus Publications remains neutral with regard to jurisdictional claims in published maps and
institutional affiliations.
**Acknowledgments**

This study is supported by the Key Research and Development Program of Zhejiang Province

(2021C03017), the National Key Research and Development Program of China (2021YFD1700802) and
the Natural Science Foundation of Zhejiang Province (LY23E090001). Taizhou Municipal Bureau of
Water Resources is greatly acknowledged for providing meteorological and hydrological data used in the



study area.

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
