# Peer review of "Synchronization frequency analysis and stochastic"

_EGUsphere, 2024_

## Author Response (AR1)

**Reply to the Editors and the Reviewers' comments on Manuscript EGUSPHERE-2024-2266**

Dear Editors and Reviewers,

Thank you very much for your evaluation of our manuscript and insightful comments, which have been a great help in improving the quality of our manuscript. We have carefully revised the manuscript according to your comments and suggestions. The related parts of the manuscript have been rewritten and improved, and for your easy reading and evaluation, the changed parts are marked using track changes text in the revised version.

We hope that the modifications introduced are satisfactory and the paper is now suitable for publication in *Hydrology and Earth System Sciences*. Below we list how the specific comments of editors and reviewers are addressed.

**Responses to referee #1**

**1.1. Subsection 2.3.1: In this subsection, the description of the text and the presentation of Figure 3 focus on illustrating the process of how to choose the key variables. But after selecting the key variables, how to construct the RDV-Copula model needs further elaboration. Please supplement this section and modify the picture if necessary.**

**Reply to 1.1:** Thank you for your suggestion. Figure 3 has been modified. In addition, the manuscript has been supplemented with the entirety of the RDV-Copula model construction. Please see Line 282-293, Page 12-13.

[Figure]

**Figure 4. Schematic diagram of the RDV-Copula method**

"*After identifying the "N+1" key variables, the marginal distribution function for each variable is determined, selecting the most appropriate distribution (e.g., Normal, Gamma) based on the statistical characteristics of each variable. Using these marginal distributions, a suitable copula structure is then selected, such as C-Vine or D-Vine, depending on the nature of dependencies among the key variables. Next, for each pair of variables in the chosen vine structure, the most appropriate bivariate copula family (e.g., Gaussian, Clayton, Gumbel) is selected to accurately capture their dependencies. Subsequently, parameters for each selected pair-copula are estimated sequentially using methods like Maximum Likelihood Estimation (MLE). Finally, the constructed copula model is validated using statistical criteria such as the Akaike Information Criterion (AIC).*"

**1.2. Subsection 3.2.2.2: Why are these three different sets of structures chosen as benchmark models? What is the significance of the comparison of each set of benchmarks? Please explain how is it possible to validate the effectiveness of the proposed RDV-Copula method by comparing it with the three sets of benchmarks?**

**Reply to 1.2:** The three different benchmark models were chosen to evaluate the RDV-Copula method across distinct dimensions of complexity and correlation (spatial and temporal) to assess its effectiveness in capturing dependencies. Here's why each set of structures was selected and their significance in validating the proposed RDV-Copula method:

**Benchmark 1: Four-dimensional spatial vine copula (Unconditional)**

- **Reason for selection:** This benchmark focuses solely on the effect of spatial correlations, providing a simpler case where only inflows from the four sites (LSM-LX-QS-SD) on the same day are considered.

- **Significance:** By limiting the model to spatial correlations, Benchmark 1 provides a baseline to compare how well the RDV-Copula captures spatial dependencies. If the RDV-Copula outperforms this benchmark, it demonstrates that including temporal correlations (as RDV-Copula does) improves performance.

**Benchmark 2: Eight-dimensional spatial-temporal vine copula (Unconditional)**

- **Reason for selection:** This benchmark extends the analysis by incorporating both spatial and temporal correlations. The inclusion of inflows from both the current and previous day (LSM-LX-QS-SD-LSM1-LX1-QS1-SD1) reflects a more complex dependence structure.

- **Significance:** This model demonstrates the performance when handling both spatial and temporal correlations in an unconditional framework. Comparing it with RDV-Copula shows whether the latter's reduced dimensionality (five-dimensional) or conditional simulation better captures the hydrological dynamics.

**Benchmark 3: Eight-dimensional spatial-temporal vine copula (Conditional)**

- **Reason for selection:** This benchmark uses a similar eight-dimensional

structure as Benchmark 2 but incorporates conditional simulation, assuming that the previous day's runoff is known.

- ◆ **Significance:** By comparing it with RDV-con (which also uses conditional simulation but with a simplified five-dimensional structure), the comparison highlights whether the RDV-Copula's dimensional reduction sacrifices accuracy or remains effective.

**How to validate the effectiveness of the proposed RDV-Copula method:**

- ◆ **Comparing RDV-un with Benchmark 1:** If RDV-un (which includes both spatial and temporal correlations) outperforms Benchmark 1 (spatial-only), it validates that considering temporal information adds value.

- ◆ **Comparing RDV-un with Benchmark 2:** A comparison with Benchmark 2 (eight-dimensional) demonstrates whether RDV-un's reduced dimensionality preserves or enhances predictive performance, thereby evaluating the RDV-Copula's ability to balance model complexity and accuracy.

- ◆ **Comparing RDV-con with Benchmark 3:** RDV-con's conditional simulation can be compared with Benchmark 3's approach to assess whether the reduced dimensionality of the RDV-Copula still captures the essential conditional temporal dynamics.

Overall, the comparison with these benchmarks allows for an assessment of whether the RDV-Copula method is both effective in reducing model complexity and capturing critical relationships, validating its practicality for applications.

**1.3. Line 418-421: What does the symbol of the "*" in Figure 7(a) indicate? There is no explanation in the text or in the picture.**

**Reply to 1.3:** Sorry for confusion. The manuscript has been revised by adding the following sentence to explain the symbol "*". Please see Line 469, Page 20.

*"The "*" on the ellipse means that the correlation passes the significance test of $\alpha = 0.05$."*

**1.4. Line 198-201: The methodology provides a brief introduction to the**

**differences and characteristics of C-vine and D-vine copula. However, there is still a possibility that it may confuse readers who do not have the knowledge of this area. I recommend that some schematic diagrams may be added to the introduction to assist comprehension.**

**Reply to 1.4:** Thanks for your suggestion. C-vine and D-vine have different structures. The schematic diagrams of these two are shown below. In the C-vine copula structure, each tree features a central node that is connected to all other edges, as illustrated in Figure 2(a). C-vine is suitable for structures with a key variable that has a significant correlation with the remaining other variables. In contrast, in the D-vine copula structure, each node is connected to no more than two edges, as depicted in Figure 2(b). The set of diagrams and a more detailed description have been added to the manuscript. Please see Line 200-207, Page 8.

*"In the C-vine copula structure, each tree features a central node that is connected to all other edges, as illustrated in Figure 2(a). C-vine is suitable for structures with a key variable that has a significant correlation with the remaining other variables. In contrast, in the D-vine copula structure, each node is connected to no more than two edges, as depicted in Figure 2(b). The order of dependencies between variables can be determined by one after the other. The expressions for the n-dimensional joint probability density of C-vine and D-vine are shown in Equations (4) and (5)."*

[Figure]

(a)                                                    (b)

**Figure 2. The vine structures for the given order of 3 variables in (a) the C-vine copula and (b) the D-vine copula**

**1.5. In the process of constructing the joint distribution function, why is only the relationship between yesterday's runoff and today's runoff considered when identifying the relationship in the time dimension? Why not consider the effect of the runoff from two days ago on today's runoff?**

**Reply to 1.5:** Sorry for confusion. The reason is that the magnitude of the degree of association needs to be considered when identifying key variables in the temporal dimension used to construct the joint distribution function. Take the runoff of LSM site as an example. LSM represents the runoff data of the current day, LSM1 represents the runoff data of the previous day, LSM2 represents the runoff data of two days ago, LSM3 represents the runoff data of three days ago, and LSM4 represents the runoff data of four days ago. The correlations between LSM and LSM1, LSM2, LSM3, and LSM4 are represented by (a)-(d) in the figure below. Pearson's correlation coefficients show a gradual decrease in correlation with time. The correlation is highest for two adjacent days.

[Figure]

Here are the Pearson correlation coefficients:

- LSM and LSM1: 0.566

- LSM and LSM2: 0.311

- LSM and LSM3: 0.204

- LSM and LSM4: 0.180

It can be concluded that the correlation between LSM and the previous day's runoff is the highest with 0.566. While the data from two days ago no longer has much influence on the current day's runoff data, so it can be excluded from the critical variable selection. Considering only the previous day's contribution in the time dimension can effectively represent the time correlation while avoiding unnecessary dimension increase. This has been modified in the manuscript. Please see Line 408-418, Page18.

*"Given that all four sites are situated within the Shifeng Creek watershed, their spatial interconnectivity is inherent and can be leveraged in constructing a vine copula function. Additionally, the results of the correlation analysis indicate that the correlation between the current day's runoff and the previous day's runoff is the highest. While the data from two days ago no longer has much influence on the current day's runoff data, so it can be excluded from the critical variable selection. Considering only the previous day's contribution in the time dimension can effectively represent the time correlation while avoiding unnecessary dimension increase. This study integrates the inflows from the four sites over two consecutive days. The inflows for the current day are denoted as LSM, LX, QS, and SD, while those for the previous day are labeled LSM1, LX1, QS1, and SD1, respectively."*

**1.6. Line 378-379: "This chosen structure is then further compared with other copula functions to validate its efficacy." Based on my understanding, the phrase "*This chosen structure*" refers to the structure selected after comparing the RDV-Cvine and RDV-Dvine. However, the sentence is somewhat ambiguous, so I am uncertain if my interpretation is correct. Could you please clarify and rewrite the sentence?**

**Reply to 1.6:** Sorry for causing such confusion. The vine copula structure (RDV-Cvine

or RDV-Dvine) with better index values will be preferred. "*This chosen structure*" refers to the structure selected after comparing the RDV-Cvine and RDV-Dvine. The sentence has been modified. Please see Line 433-434, Page 19.

*"The RDV-Copula structure with better index values is then further compared with other copula functions to validate its efficacy."*

**1.7. I think there is a repetition of the information presented in the table and the picture in Supplement D. Please select one of them (table or picture) for the information presentation.**

**Reply to 1.7:** Thanks for your suggestion. In Supplement D, only the pictures of the marginal distribution function are preserved for display. The table content has been removed.

**1.8. Line 98-100: "This complexity can complicate the copula's structure determination, inflate computational demands during parameter fitting, and potentially diminish the accuracy of stochastic simulations." In this sentence, the phrase "copula's structure determination" should be revised to "copula structure's determination".**

**Reply to 1.8:** Thanks for your advice. This has been modified in the manuscript. Please see Line 99-100, Page 4.

*"This complexity can complicate the copula structure's determination, inflate computational demands during parameter fitting, and potentially diminish the accuracy of stochastic simulations."*

**1.9. Line 365-367: "The subsequent step involves identifying the site with the most significant correlation to its preceding day's inflow, which is then used as a as a variable to represent the temporal relationship on that day." There is an error in this sentence. "as a" is repeated, please delete the redundant one.**

**Reply to 1.9:** Thanks for your suggestion. This has been modified in the manuscript. Please see Line 420-422, Page 18.

*"The subsequent step involves identifying the site with the most significant correlation to its preceding day's inflow, which is then used as a variable to represent the temporal relationship on that day."*

**1.10. Line 442-443: "The obvious dark colored blocks in the graph indicate the high probabilities of being the high-water or the low-water concurrently." This sentence seems a bit confused. Please rewrite it to avoid ambiguity.**

**Reply to 1.10:** Sorry for causing such confusion. This sentence has been revised in the manuscript. Please see Line 499-500, Page 23.

*"The obvious dark-colored blocks in the graph indicate the high probabilities of being in high-water or low-water states concurrently."*

**1.11. Line 445-447: "While the LSM site's synchronization probabilities with the other sites are comparatively lower, they still exceed 50%, recorded at 58.29% with the LX site, 61.25% with the QS site, and 57.15% with the SD site." The sentence is not clear enough, please revise it and replace the word "recorded".**

**Reply to 1.11:** Thanks for your advice. This sentence has been revised in the manuscript. Please see Line 505-507, Page 23.

*"While the LSM site's synchronization probabilities with the other sites are comparatively lower, they still exceed 50%, with values of 58.29% for the LX site, 61.25% for the QS site, and 57.15% for the SD site."*

**1.12. Line 653-655: "Depending on the number of hydrometric stations, Wang and Shen (2023b) established the 7-dimensional regular vine (R-vine) copula models to depict the complex and diverse dependence." Please delete the "the" before "7-dimensional regular vine". Please replace the "dependence" by "dependencies".**

**Reply to 1.12:** Thanks for your advice. This has been modified in the manuscript. Please see Line 714-716, Page 35.

*"Depending on the number of hydrometric stations, Wang and Shen (2023b) established 7-dimensional regular vine (R-vine) copula models to depict the complex*

*and diverse dependencies.”*

**1.13. Line 700: “The conditional simulation is a double-edged sword.” Please remove the “the” before the conditional simulation.**

**Reply to 1.13:** Thanks for your advice. This has been modified in the manuscript. Please see Line 763, Page 37.

> *“Conditional simulation is a double-edged sword.”*

**References#1**

Wang, X., & Shen, Y.-M. (2023). A Framework of Dependence Modeling and Evaluation System for Compound Flood Events. *Water Resources Research*, *59*(8), e2023WR034718. https://doi.org/10.1029/2023WR034718

**Responses to referee #2**

**\* Introduction:**

It is generally well-written, with good structure and clarity, but there are a few areas where improvements can be made for better precision and readability.

**2.1. \* Line 31-33: "As is reported by Centre for Research on the Epidemiology of Disasters (CRED)" The use of "is" here is unnecessary and makes the sentence sound awkward. Remove the word "is" from the sentence.**

**Reply to 2.1:** Thanks for your suggestion. The word "is" has been removed from this sentence. This was also modified in the manuscript. Please see Line 31-32, Page 2.

*"As reported by Centre for Research on the Epidemiology of Disasters (CRED), floods represented 45.6% of worldwide natural disasters in 2022, affecting an average of 57.1 million people annually (CRED,2023)."*

**2.2. \* Line 37: "Large floods often result from the amalgamation of floods from multiple sub-watersheds" Please replace "amalgamation" with "merging" for more concise language.**

**Reply to 2.2:** Thanks for your suggestion. This has been modified in the manuscript. Please see Line 37-38, Page 2.

*"Large floods often result from the merging of floods from multiple sub-watersheds*(Prohaska & Ilic, 2010)*."*

**2.3. \* Line 64-65: "Copula function is widely applied in hydrological fields..." "Copula function" should be plural here.**

**Reply to 2.3:** Thanks for your suggestion. This has been revised in the manuscript. Please see Line 65-70, Page 3.

*"Copula functions are widely applied in hydrological fields, including the joint frequency analysis (Liu et al., 2018; B. Zhang et al., 2021), water resources management (X. Gao et al., 2018; Nazeri Tahroudi et al., 2022), wetness-dryness*

*encountering (S. Wang et al., 2022; S. Zhang et al., 2023), flood risk assessment (Li et al., 2022; Tosunoglu et al., 2020; Zhong et al., 2021) , water quality analysis (Yu et al., 2020; Yu & Zhang, 2021),  precipitation model (C. Gao et al., 2020; Nazeri Tahroudi et al., 2023; Tahroudi et al., 2022) and so on."*

**\* Method:**

**2.4. \* Many equations are presented in the paper, and most look OK. However, please check carefully whether all equations are necessary and whether the quantities involved are properly explained. Line 232-237: For equations (6), (7), and (8), the variables are not clearly explained. Please elaborate on the meaning of each variable, such as $u_{ph}^1, u_{ph}^2, u_{ph}^3, u_{ph}^4$ and $u_{pl}^1, u_{pl}^2, u_{pl}^3, u_{pl}^4$. This ensures the reader understands the variables used in the equations.**

**Reply to 2.4:** Sorry for causing such confusion. Since the inflows of the different sites are represented by $X^1, X^2, X^3, \cdots, X^{N+M}$. The variables $X_{ph}^1, X_{ph}^2, X_{ph}^3, \cdots, X_{ph}^{N+M}$ represent the inflow amounts corresponding to the high-water at these sites, while $X_{pl}^1, X_{pl}^2, X_{pl}^3, \cdots, X_{pl}^{N+M}$ denote the inflow amounts corresponding to the low-water at the respective sites. The marginal distribution functions for these inflows are represented as $u^1, u^2, u^3, \cdots, u^{N+M}$. Specifically, $u_{ph}^1, u_{ph}^2, u_{ph}^3, \cdots, u_{ph}^{N+M}$ denote the marginal distribution functions corresponding to the high-water inflow amounts $X_{ph}^1, X_{ph}^2, X_{ph}^3, \cdots, X_{ph}^{N+M}$, capturing the probabilistic behavior of the inflows during high-water conditions at each site. Similarly, $u_{pl}^1, u_{pl}^2, u_{pl}^3, \cdots, u_{pl}^{N+M}$ represent the marginal distribution functions for the low-water inflow amounts $X_{pl}^1, X_{pl}^2, X_{pl}^3, \cdots, X_{pl}^{N+M}$, describing the inflow behavior during low-water conditions at these sites. The explanation of this part has been supplemented in the manuscript. Please see Line 229-237, Page 10.

*"Let the inflows of the different sites be represented by $X^1, X^2, X^3, \cdots, X^{N+M}$. $X_{ph}^1, X_{ph}^2, X_{ph}^3, \cdots, X_{ph}^{N+M}$ represent the amounts of inflow corresponding to the high-*

*water of these different sites respectively. Meanwhile,* $X_{pl}^1, X_{pl}^2, X_{pl}^3, \cdots, X_{pl}^{N+M}$

*represent the amounts of inflow corresponding to the low-water of these different sites*

*respectively. The marginal distribution functions are* $u^1, u^2, u^3, \cdots, u^{N+M}$, *respectively.*

*Specifically,* $u_{ph}^1, u_{ph}^2, u_{ph}^3, \cdots, u_{ph}^{N+M}$ *denote the marginal distribution functions*

*corresponding to the high-water inflow amounts* $X_{ph}^1, X_{ph}^2, X_{ph}^3, \cdots, X_{ph}^{N+M}$, *capturing*

*the probabilistic behavior of the inflows during high-water conditions at each site.*

*Similarly,* $u_{pl}^1, u_{pl}^2, u_{pl}^3, \cdots, u_{pl}^{N+M}$ *represent the marginal distribution functions for the*

*low-water inflow amounts* $X_{pl}^1, X_{pl}^2, X_{pl}^3, \cdots, X_{pl}^{N+M}$, *describing the inflow behavior*

*during low-water conditions at these sites."*

**2.5. * Figure2. The colors used to represent elements such as reservoirs and cross sections are a bit confusing. Use the same color for the same elements throughout the figure. For instance, if reservoirs are represented by a specific color, maintain that color consistently. Please revise it.**

**Reply to 2.5:** Sorry for the confusion. The figure has been modified, as shown below. The colors of various elements in the figure were unified. The reservoirs are shown in green and the cross-sections are shown in blue.

[Figure]

**Figure 2. Schematic diagram of the generalized system in the catchment**

**2.6. * Figure4. How is it possible to distinguish between conditional simulation and unconditional simulation through Figure 4? There is no clear explanation or**

**visual distinction between conditional and unconditional simulation in the figure. Please provide a clear legend or add explanations for (a) and (b) in Figure 4, highlighting the difference between conditional and unconditional simulations.**

**Reply to 2.6:** Thank you for your suggestion. Figure 4 has been modified. The descriptions for both conditional and unconditional simulations have been further refined and supplemented to enhance clarity. This has been revised in the manuscript. Please see Line 302-324, Page 13-14.

*"Unconditional simulation (Figure 5(a)): This approach generates random samples based solely on the marginal probability distribution, without incorporating any existing data constraints. The probability distribution is shown in the upper-left plot, and random samples are generated simultaneously, resulting in the scatter plot below. The generated samples, represented by blue points, illustrate the joint variability according to their predefined marginal distributions. Since no prior information is used, each data point is in an unknown state before the simulation.*

*Conditional simulation (Figure 5(b)): In this scenario, the simulation takes into account pre-existing data conditions. The marginal probability distribution is displayed in the top-center plot, while the known conditional data is shown in the upper-right scatter plot (in pink). These known data points act as a constraint for generating new random samples. The resulting scatter plot below (blue and pink points) demonstrates how the conditional samples are influenced by both the marginal distribution and the specified conditions of the known data. This method allows for a tailored simulation that incorporates pre-existing data insights."*

[Figure]

**Figure 4. Schematic diagram for generating random simulation samples (a) unconditional simulation (b) conditional simulation**

**\* Case study:**

**2.7. \* Line 316-317: "This study focuses on four major sites within the Shifeng Creek catchment" The reason for selecting these sites is unclear. Clarify that these sites are chosen due to their strategic importance. Can you explain briefly why these particular sites were selected for the study?**

**Reply to 2.7:** These four sites were selected due to their strategic importance within the Shifeng Creek catchment, representing key locations in the upper, middle, and lower reaches of the river system. The Lishimen Reservoir (LSM) and Longxi Reservoir (LX) sites are both situated in the upper reaches of the catchment and play a crucial role in flood control by regulating inflows and managing water levels. Understanding the flow patterns at these reservoir sites is essential for optimizing reservoir operations and mitigating downstream flood risks, especially during peak flood periods. The Qianshan (QS) cross-section, located in the middle reaches, and the Shaduan (SD) cross-section, positioned in the lower reaches, are critical control points for flood management. Analyzing flow processes at these sections allows for better coordination of reservoir operations and helps prevent the convergence of flood peaks, thus enhancing flood mitigation throughout the catchment. The reasons for why these four critical sites were

chosen has been supplemented in the manuscript. Please see Line 360-366, Page 16.

*"These four sites were selected for their strategic importance within the Shifeng Creek catchment, covering the upper, middle, and lower reaches. The Lishimen (LSM) and Longxi (LX) reservoirs, both in the upper reaches, are vital for flood control, regulating inflows to reduce downstream flood risks. The Qianshan (QS) cross-section, in the middle reaches, and the Shaduan (SD) cross-section, in the lower reaches, serve as key flood control points. Analyzing flows at these sites enables better coordination of reservoir operations and prevents flood peak convergence, enhancing overall flood management."*

**2.8. * Line 344-346: The combinations of [X-H, Y-H, Z-H, W-H], [X-M, Y-M, Z-M, W-M], and [X-L, Y-L, Z-L, W-L] mean what synchronization respectively? The abbreviations are used without prior explanation. Before utilizing the abbreviations please correspond the abbreviated letters to the original meaning. If not, it will be harder for readers to understand.**

**Reply to 2.8:** Sorry for the confusion. These three combinations represent different types of synchronization. Specifically, [X-H, Y-H, Z-H, W-H] indicates that all four sites (X, Y, Z, and W) are in a high-water state, [X-M, Y-M, Z-M, W-M] signifies that all sites are in a medium-water state, and [X-L, Y-L, Z-L, W-L] means that all sites are in a low-water state simultaneously. To improve clarity, the manuscript has been revised. Please see Line 390-397, Page 17.

*"Considering the three potential states (High/Medium/Low) at each site, a total of 81 possible inflow-state combinations are identified. For ease of presentation, H, M, and L are then used as abbreviations for High, Medium, and Low. Among the 81 combinations, the combinations [X-H, Y-H, Z-H, W-H], [X-M, Y-M, Z-M, W-M], and [X-L, Y-L, Z-L, W-L] are classified as synchronous high-water, synchronous medium-water, synchronous low-water, respectively, while the remainder are deemed asynchronous. The calculation equations can be referenced in Appendix B."*

**2.9. * Line 320-321: "To achieve this, daily runoff data of August, covering a span**

**from 2000 to 2020, have been compiled.” This sentence would be more properly expressed in the past tense. Change "have been compiled" to "were compiled" for better tense consistency.**

**Reply to 2.9:** Thanks for your suggestion. This has been modified in the manuscript. Please see Line 366-367, Page 16.

*"To achieve this, daily runoff data of August, covering a span from 2000 to 2020, were compiled."*

**\* Results**

**2.10. \* This section is well written. The only area of weakness is the display of pictures. The font size of figure 10 is a little small. It could be considered to keep only a lower number of sub-figures and the rest could be placed in supplementary part of the article.**

**Reply to 2.10:** Thanks for your constructive suggestion. The font size in the figure has been enlarged. Here are the modified figures. Moreover, to ensure the clarity of each figure, only 9 figures are put here, and the rest are put in the supplemental content.

[Figure]

**Figure 10.   Cumulative probability distribution of the preferred marginal distribution function for runoff on each day throughout 1st-9th in August**

**\* Discussion & Conclusion**

**2.11. \* Line 666-668: "Although the eight-dimensional vine copula model takes more variables into account, including both temporal and spatial correlation, the model is too complicated due to many variables, which makes the simulation less efficient on the contrary." This sentence is too long and complex. Pease simplify it.**

**Reply to 2.11:** Thanks for your constructive suggestion. This sentence has been modified in the manuscript. Please see Line 729-731, Page 36.

*"Although the eight-dimensional vine copula model considers both temporal and spatial correlations, its complexity reduces simulation efficiency due to the large number of variables."*

**\* Minor comments**

The grammar in the article is generally correct. However, there are some words that are not used appropriately. A few examples are given below:

**2.12. \* Line 254-255: "This strategy aims to distill essential spatial-temporal information, thereby reducing the vine copula function's dimensionality to simplify the model structure." Here, the meaning of the word "distill" is not suitable for this application. It would be more appropriate to replace it with "extract".**

**Reply to 2.12:** Thanks for your suggestion. The word "distill" has been replaced. And this sentence has been modified in the manuscript. Please see Line 267-268, Page 11.

*"This strategy aims to extract essential spatial-temporal information, thereby reducing the vine copula function's dimensionality to simplify the model structure."*

**2.13. \* Line 271-273: "The core distinction between these two simulation methods hinges on whether certain data points are pre-determined at the time of simulation." There is something wrong with the logic of this statement.**

**Reply to 2.13:** Sorry for the confusion. This sentence has been modified in the manuscript. Please see Line 299-300, Page 13.

*"The key difference between these two simulation methods lies in whether specific data points are known in advance before generating the simulation."*

**2.14. * Line 347: "The calculation equations can be referenced in Appendix B." I think "provided" may be better than "referenced".**

**Reply to 2.14:** Thanks for your suggestion. The word "referenced" has been replaced by "provided". And this sentence has been modified in the manuscript. Please see Line 396-397, Page 17.

*"The calculation equations can be provided in Appendix B."*

**2.15. * The supplemental content section is a bit too redundant. Think about keeping the important parts.**

**Reply to 2.15:** Thanks for your suggestion. Appendix A, B, and C are all essential. The table in Appendix D has been removed and only the figures of the marginal distribution preference results are retained for presentation.

**References#2**

Gao, C., Booij, M. J., & Xu, Y.-P. (2020). Development and hydrometeorological evaluation of a new stochastic daily rainfall model: Coupling Markov chain with rainfall event model. Journal of Hydrology, 589, 125337. https://doi.org/10.1016/j.jhydrol.2020.125337

Gao, X., Liu, Y., & Sun, B. (2018). Water shortage risk assessment considering large-scale regional transfers: a copula-based uncertainty case study in Lunan, China. Environmental Science and Pollution Research, 25(23), 23328–23341. https://doi.org/10.1007/s11356-018-2408-1

Li, R., Xiong, L., Jiang, C., Li, W., & Liu, C. (2022). Quantifying multivariate flood risk under nonstationary condition. Natural Hazards. https://doi.org/10.1007/s11069-022-05716-x

Liu, Z., Cheng, L., Hao, Z., Li, J., Thorstensen, A., & Gao, H. (2018). A Framework for Exploring Joint Effects of Conditional Factors on Compound Floods. Water Resources Research, 54(4), 2681–2696. https://doi.org/10.1002/2017WR021662

Nazeri Tahroudi, M., Ramezani, Y., De Michele, C., & Mirabbasi, R. (2022). Trivariate joint frequency analysis of water resources deficiency signatures using vine copulas. Applied Water Science, 12(4), 67. https://doi.org/10.1007/s13201-022-01589-4

Nazeri Tahroudi, M., Ahmadi, F., & Mirabbasi, R. (2023). Performance comparison of IHACRES, random forest and copula-based models in rainfall-runoff simulation. Applied Water Science, 13(6), 134. https://doi.org/10.1007/s13201-023-01929-y

Prohaska, S., & Ilic, A. (2010). Coincidence of Flood Flow of the Danube River and Its Tributaries. In M. Brilly (Ed.), Hydrological Processes of the Danube River Basin: Perspectives from

the Danubian Countries (pp. 175–226). Dordrecht: Springer Netherlands. https://doi.org/10.1007/978-90-481-3423-6_6

Tahroudi, M. N., Mohammadi, M., & Khalili, K. (2022). The application of the hybrid copula-GARCH approach in the simulation of extreme discharge values. Applied Water Science, 12(12), 274. https://doi.org/10.1007/s13201-022-01788-z

Tosunoglu, F., Gurbuz, F., & Ispirli, M. N. (2020). Multivariate modeling of flood characteristics using Vine copulas. Environmental Earth Sciences, 79(19), 459. https://doi.org/10.1007/s12665-020-09199-6

Wang, S., Zhong, P.-A., Zhu, F., Xu, C., Wang, Y., & Liu, W. (2022). Analysis and Forecasting of Wetness-Dryness Encountering of a Multi-Water System Based on a Vine Copula Function-Bayesian Network. Water, 14(11), 1701. https://doi.org/10.3390/w14111701

Yu, R., & Zhang, C. (2021). Early warning of water quality degradation: A copula-based Bayesian network model for highly efficient water quality risk assessment. Journal of Environmental Management, 292, 112749. https://doi.org/10.1016/j.jenvman.2021.112749

Yu, R., Yang, R., Zhang, C., Spoljar, M., Kuczynska-Kippen, N., & Sang, G. (2020). A Vine Copula-Based Modeling for Identification of Multivariate Water Pollution Risk in an Interconnected River System Network. Water, 12(10), 2741. https://doi.org/10.3390/w12102741

Zhang, B., Wang, S., & Wang, Y. (2021). Probabilistic Projections of Multidimensional Flood Risks at a Convection-Permitting Scale. Water Resources Research, 57(1). https://doi.org/10.1029/2020WR028582

Zhang, S., Kang, Y., Gao, X., Chen, P., Cheng, X., Song, S., & Li, L. (2023). Optimal reservoir operation and risk analysis of agriculture water supply considering encounter uncertainty of precipitation in irrigation area and runoff from upstream. Agricultural Water Management, 277, 108091. https://doi.org/10.1016/j.agwat.2022.108091

Zhong, M., Zeng, T., Jiang, T., Wu, H., Chen, X., & Hong, Y. (2021). A Copula-Based Multivariate Probability Analysis for Flash Flood Risk under the Compound Effect of Soil Moisture and Rainfall. Water Resources Management, 35(1), 83–98. https://doi.org/10.1007/s11269-020-02709-y